# New Developments in Incremental Heating Detrital $^{40}$Ar/$^{39}$Ar Lithic (DARL) Geochronology using Icelandic River Sand

Odinaka Okwueze[1], Kevin Konrad[1,2]*, Tomas Capaldi[1,3]

[1] Department of Geoscience, University of Nevada Las Vegas, Las Vegas, Nevada 89154, USA

[2] Now at College of Earth, Ocean and Atmospheric Sciences, Oregon State University, Corvallis, OR 97331, USA

[3] Now at Scripps Institution of Oceanography, University of California San Diego, La Jolla, California 92093, USA

*Correspondence to*: Kevin Konrad ([Kevin.Konrad@oregonstate.edu](mailto:Kevin.Konrad@oregonstate.edu)),

**Abstract.** Iceland records over fifteen million years of complex volcanism resulting from the intersection of mid-ocean ridge and mantle plume upwelling. The Iceland mantle plume has been active for at least 70 Ma, with surface expressions in
Greenland, the North Atlantic, and Iceland. The Iceland hotspot may exhibit periods of increased volcanic output linked to pulses of upwelling within the plume. Understanding Iceland's magmatic history and potential pxulsation could provide key insights into dynamic topography driving changes in deep-water oceanic circulation, late Cenozoic climate change and mantle plume - mid-ocean ridge interaction. Detrital geochronology is a powerful tool for capturing the magmatic history of a region. However, Iceland's fine-grained extrusive volcanic lithologies lack the typical detrital mineral phases such as zircon, sanidine,
hornblende, and rutile that current geochronology methods utilize. Here we present a new methodology for capturing the magmatic history of fine grained extrusive volcanic rocks using single grain detrital $^{40}$Ar/$^{39}$Ar incremental heating geochronology. The DARL (or Detrital Argon Lithics) method has consisted of $^{40}$Ar/$^{39}$Ar incremental heating and total fusion analyses on single lithic grains, which has not yet been applied to predominantly mafic terrains composed of young glassy lava flows, which commonly display subatmospheric $^{40}$Ar/$^{36}$Ar isochron intercepts and low $^{40}$Ar*. This work represents a $^{40}$Ar/$^{39}$Ar
incremental heating pilot study on nineteen single grains of Icelandic river sand/fine gravel (1-3 mm), collected from five different catchments. Fifteen of the nineteen basaltic grains produced concordant age experiments that ranged from 0.2 to 13.5 Ma and uncertainties (2σ) from 1% to 86% with the grains under 1 Ma having the largest uncertainties. Preliminary results show that basaltic grains with less alteration (and corresponding lower atmospheric argon concentration) yield more accurate age determinations, though altered basaltic grains can still produce statistically valid age determinations. Results presented
here show the validity of the incremental heating DARL methodology for capturing the magmatic history of mafic terrains. The long analysis time required for incremental heating experiments makes it infeasible to produce the large number of ages required for a detrital study. For this reason, we build upon a previously proposed method that combines total fusion and incremental heating DARL methodologies to acquire age data for the large N values needed for detrital studies of mafic volcanic terrains.

# 1 Introduction

Iceland is a unique ocean island that represents the long-term interaction between a mantle plume and mid-ocean ridge (Figure 1; Morgan, 1971; Gudmundsson, 2000; Harðarson et al., 2008). The island has been volcanically active for at least 15 m.y. and potentially displays temporal pulsation in melt output (e.g. O'Connor et al., 2000; Parkin et al., 2007; Rychert et al., 2018). Understanding the variations in melt output through time is vital for constraining mantle plume dynamics, deep Earth – climate interactions, and dynamic topography drivers. However, determining the ages of all exposed lithologies is financially improbable and previous efforts to obtain a detrital geochronologic dataset of the region has been limited by a lack in variation among zircon U-Pb ages (Carley et al., 2017). This limitation is due to the relatively low volumes of felsic volcanism on the island. Instead, fine grained lava flows, which do not host minerals commonly used for detrital studies such as zircon and rutile, dominate the surface geology. Therefore, an alternative methodology is required to constrain the detrital geochronologic history of the region.

The detrital $^{40}Ar/^{39}Ar$ lithic (DARL) method is a relatively new detrital geochronological tool that determines the $^{40}Ar/^{39}Ar$ total fusion or incremental heating age determinations on single grains or multi-grain aliquots recovered from sedimentary deposits (e.g. watersheds) (Benowitz et al., 2014, 2018; Vanderleest et al., 2020; Trop et al., 2022; Kenny et al., 2022). The technique was first reported by Benowitz et al. (2014), wherein incremental heating analyses were undertaken on fine-grained volcanic lithics to propose refined total fusion temperature ranges for rapid DARL analyses. The DARL method was employed to determine the history of the Wrangell Volcanic Arc (Alaska, USA) through 2771 analyses of grains, ranging in size from sand to cobble (Trop et al., 2022). The DARL ages matched the expected age range based on available bedrock analyses (Trop et al., 2022; Brueseke et al., 2023). The chemistry and age results from this technique allowed for novel insights into the evolution of the Wrangell Arc that were only partially observed using traditional U-Pb detrital zircon analyses (Trop et al., 2022; Brueseke et al. 2023). Similarly, Vanderleest et al. (2020) performed incremental heating experiments on igneous clasts separated from a conglomeratic formation (n=7), which provided detrital chronologic constraints on the evolution of the Magallanes-Austral basin within the southern Patagonian Andes. More recently, Kenny et al. (2023) employed the DARL method on 50 sand-sized grains collected from the drainage basin of the sub-glacial Hiawatha impact structure in Greenland. Although none of the grains produced traditionally concordant heating spectrum (e.g. >50% of $^{39}Ar$ released with more than five consecutive steps), two mini-plateau ages matched resetting ages for detrital zircon. The DARL method has potential limitations due to the lower closure temperatures of Ar and greater susceptibility of age disturbances due to alteration as compared to the detrital zircon method. However, in environments that contained mixed mafic and felsic lithologies (e.g. volcanic arcs) or consist primarily of fine-grain extrusive volcanics (e.g. Iceland), the DARL method allows for novel insights not obtainable by the traditional detrital mineral phases. Here we expand upon the method through incremental heating experiments on single coarse sand or fine gravel grains of volcanic lithic fragments from Icelandic rivers. These sedimentary

deposits primarily consist of glassy or fine-grained low-K mafic lava flows and if ages can be reliably constrained with the DARL method, then other low-zircon fertility terrains such as arc and intraplate ocean islands can be constrained. Based on the incremental heating results we propose a methodology for rapid fusion analyses of glass-rich volcanic lithics.

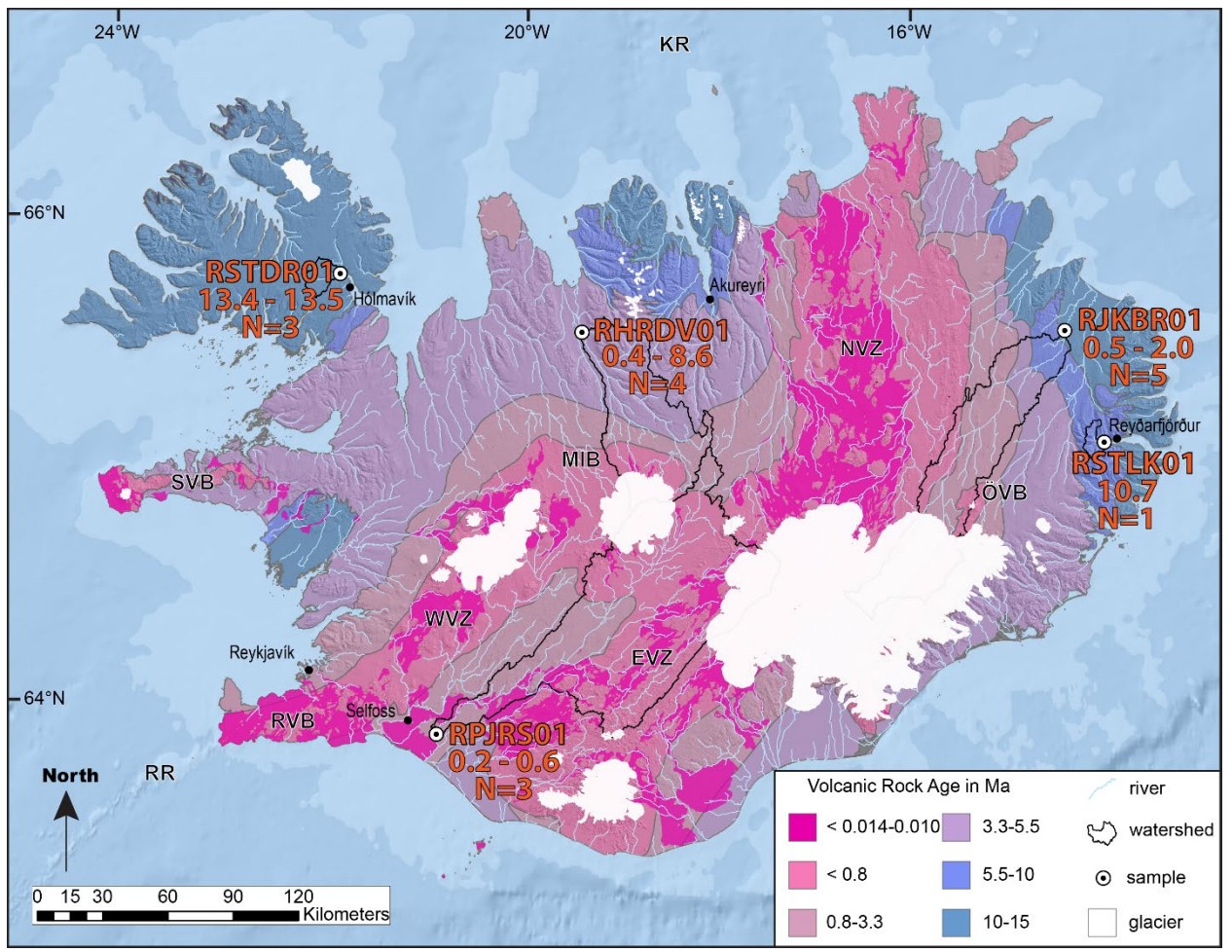

**Figure 1:** A simplified geologic map of Iceland illustrating bedrock age based on tectonic map of Jóhannesson and Sæmundsson (2009) with sample locations (white circles) and associated watersheds (black outline) overlain on the Island's DEM v1.0 (Porter et al., 2018). EVZ: East Volcanic Zone; KR: Kolbeinsey Ridge; MIB: Mid-Iceland Belt; ÖVB: Öræfi

Volcanic Belt; RR: Reykjanes Ridge; RVB: Reykjanes Volcanic Belt; SVB: Snæfellsnes Volcanic Belt; WVZ: West Volcanic Zone. Sample sites from this study are denoted with white circles. The age range (in Ma) along with number of samples attempted (N) from this study is shown.

## 2 Material and Methods

### 2.1 Sample Locations

This study incorporates bulk sediment samples recovered from the banks of five different river drainage systems in Iceland (Figure 1, Table 1). With limited numbers of strategically collected samples, modern river sediment can be used to efficiently sample large catchment areas of mafic volcanic sources to help assess the timing and pattern of magmatism across the eroding landscape of Iceland. We sampled from active bedforms along the main trunks of each river, avoiding recent disturbances and anthropogenic influx of sediment. From these five locations, 19 individual volcanic lithic grain samples were

separated for description and analysis (Figure 2). Sample RSTDRO1 was collected from the Stadará River that is within the oldest (10-15 Ma) volcanic units. The RHRDVO1 grain samples were sourced from the Heradsvötn River — within the asymmetric U-shaped valley flanking Blönduhlíðarfjöll that drains 3.3-5.5 Ma and <0.8 Ma aged volcanic rock. The RJKBRO1 samples were sourced from the Jökulsá á Brú River from a sandy channel within a gravel bar complex that erodes variable volcanic sources that span <0.8 to 10 Ma. The RSTLKO1 samples were collected from the Störilækur River, which is a

relatively small catchment area that only erodes middle Miocene volcanic rocks (5.5-10 Ma). Samples RPJRSO1 were sourced from the Pjórsá River, which drains young <0.014 to 0.8 Ma lava flows near the Mid-Iceland Belt Region.

| Sample Location ID | Latitude (°N) | Longitude (°W) | River System | General Morphology | Approximate Age Range |
|---|---|---|---|---|---|
| RSTDR01 | 65.7694 | -21.7936 | Stadará River | Gravel Bar Sands | 15-10 Ma |
| RHRDV01 | 65.5512 | -19.3636 | Heradsvötn River | Sandy dune-ripple point bar over gravel bar | 6-0 Ma |
| RJKBR01 | 65.4929 | -14.5462 | Jökulsá á Brú River | Small sandy high stage channel within large braided gravel bar system | 10-0 Ma |
| RSTLK01 | 65.0251 | -14.2567 | Störilækur River | Sandy dune-ripple point bar over gravel bar | 10-6 Ma |
| RPJRS01 | 63.8869 | -20.6992 | Pjórsá River | Various sandy lenses across large braided gravel bars channel | 0.8-0 Ma |

**Table 1:** The location and general geomorphology of each sampling site location. Age ranges are approximated from available

outcrop $^{40}$Ar/$^{39}$Ar and K/Ar age determinations collated in Jóhannesson and Sæmundsson (2009).

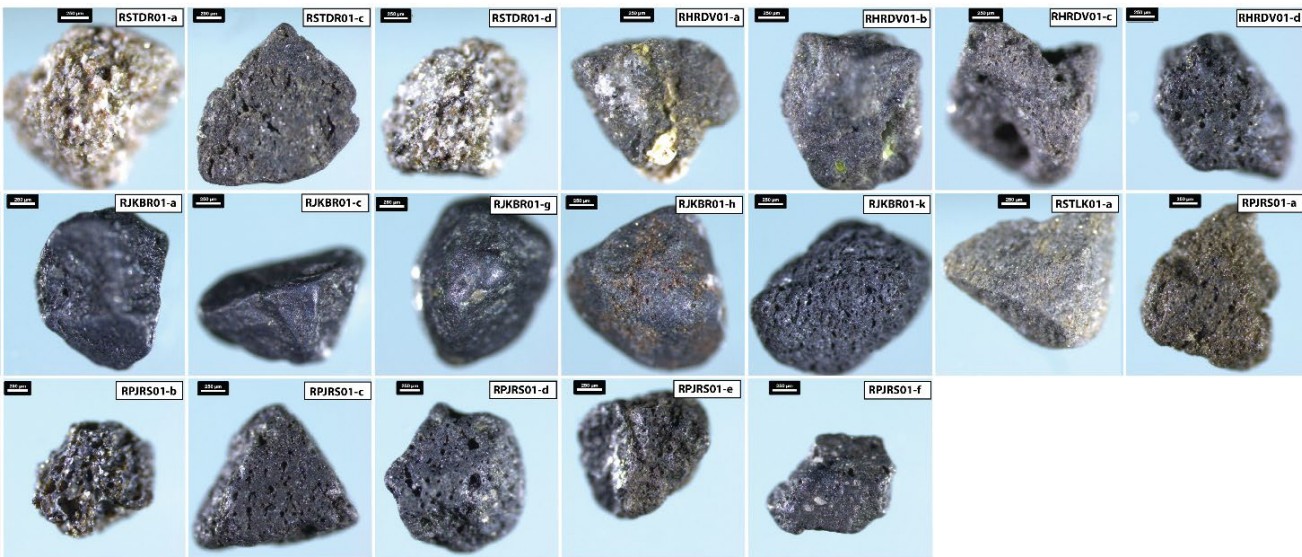

**Figure 2:** Images of the acid leached grains selected for $^{40}$Ar/$^{39}$Ar age determinations. The grains were selected to cover the general range of lithologies found in each river deposit.

## 2.2 $^{40}$Ar/$^{39}$Ar Sample Preparation

The bulk sediment samples were sieved and grains from the 2-3 mm size fraction were selected for all sites except RJKBR01, where the 1-2 mm size fraction was used. Each selected grain was separated and given a unique identifier (i.e. -A; Figure 2). The grains were photographed and described in terms of degree of crystallinity and alteration. The bulk of the samples were subjected to 30-minute sonicated baths at 50°C in 3N HCl, followed by 1N HNO$_3$, then rinsed with deionized water between each step. The only exceptions were RJKBR01-g, which was subjected to just a 15-minute acetone bath and

RKJBR01-a and -h, which were subjected to 60-minute baths in 3N HCL and then 1N HNO$_3$.

## 2.3 $^{40}$Ar/$^{39}$Ar Analysis

The individual grains were packed in Al foil and loaded into a quartz vial with Fish Canyon Tuff sanidine fluence monitors loaded at the base, top, and in between every three to four grains. The samples were sent to the TRIGA reactor at

Oregon State University to be irradiated in the cadmium-lined in-core irradiation tube (CLICIT) position for 14 hours. Once returned, samples were loaded into a high-vacuum stainless-steel extraction line at the Nevada Isotope Geochronology Laboratory and analysed via step heating in a double vacuum resistance furnace. The samples were precleaned at 400°C for 3-4 hours prior to analysis. For each step, the furnace was held at temperature for 14 minutes while the gas was exposed to a ~450°C GP-50 SAES 'getter'. The furnace was then set back to 400°C and the gas was exposed to an additional hot GP-50

getter and a room temperature GP-50 getter for six minutes to further remove reactive gases. The purified gas was then inlet

into an NGX multicollector mass spectrometer with ATONA amplifiers. The 40, 39, 38, and 37 masses were measured using amplified faraday cups while mass 36 was analysed on an ion counter. Time zero was set 20s after inlet to allow for equilibration of the gas across the mass spectrometer. Regressions consisted of 150 cycles of three second integrations with 10s off-peak baseline integrations at the start of the analysis. The data was regressed using ArArCalc software (Koppers 2002).

All age results are calibrated against a Fish Canyon Tuff age of 28.201 ± 0.023 Ma (Kuiper et al., 2008) and a total $\lambda^{40}K$ of $5.463 ± 0.107 \times 10^{-10}$ a$^{-1}$ (Min et al., 2000).

The incremental heating schedule was adjusted continually for the first four experiments (RPJRS01-b, RSTLK01-a, RJKBR01-g, RSTDR01-d) then a consistent 23 step schedule was employed for all remaining experiments. Five air standards (for mass discrimination factors; MDF) and collector calibrations (for faraday-ion multiplier calibration) were run prior to

120 every experiment. The MDF (assuming a $^{40}Ar/^{36}Ar_{atmo} = 298.56 ± 0.31$; Lee et al. 2006) and calibration factors for an individual experiment were determined by fitting a polynomial curve to the results over two weeks and interpolating the values for when the experiment was run. Neither the collector calibrations nor MDF results varied significantly over the course of the project. Furnace blanks were run prior to analyses and consisted of seven heating steps. The individual sample experiments were blank corrected by interpolation using a polynomial fit of mass concentration to temperature. We define a successful age plateau as

containing five or more consecutive heating steps that incorporate over 50% of $^{39}Ar_K$ and have a probability of fit factor >5%. If a heating step is not within uncertainty of the plateau than we refer to that as a discordant step. When a sample contained a concordant isochron with a non-atmospheric $^{40}Ar/^{36}Ar_0$ intercept (following the same statistical criteria as described for the plateau), the plateau was recalculated using the intercept and uncertainty (e.g. Heaton and Koppers, 2019). When a plateau was recalculated, no additional heating steps were added — even if they became concordant due to the increased intercept

uncertainty. All uncertainties are provided at the 2σ level and include errors on the data regression, baseline corrections, irradiation constants, the J curve value, mass fractionation, blanks, and post irradiation decay of $^{37}Ar$ and $^{39}Ar$. External uncertainties are also provided (± 2σ (f) in Table 2), which include additional propagated error from the natural element abundances, decay constant and fluence monitor age.

**3.0 Results**

Fifteen out of the nineteen experiments produced concordant age determinations (Figures 3 to 7; Table 2) with only RSTDR01-a and RHRDV01-b having any additional caveats discussed below. The ages ranged from 0.2 to 13.5 Ma, and uncertainties varied from 1 to 86% (2σ). The relative uncertainties depended primarily on the age of the sample and the corresponding $^{40}Ar*/^{40}Ar_{atm}$ corrections with younger (<1Ma) samples having highest apparent age uncertainties. Samples that

had well defined $^{36}Ar$ furnace blank fits and a high $^{40}Ar*/^{40}Ar_{atm}$ generally provided the lowest individual apparent age uncertainties. The K/Ca ($^{39}Ar_K/^{37}Ar_{Ca}$) values for the successful age plateaus range from 0.03 to 0.62, consistent with the mafic nature of the grains and corresponding terrain.

| Sample | Plateau Age | ± 2σ (i) | ± 2σ (f) | $^{39}$Ar | K/Ca | ± 2σ | MSWD | P | n | N | Inverse Isochron Age | ± 2σ (i) | ± 2σ (f) | $^{40}$Ar/$^{36}$Ar$_{int}$ | ± 2σ | SF | MSWD | P | Total Fusion Age | ± 2σ (i) | ± 2σ (f) | K/Ca |
|---|---|---|---|---|---|---|---|---|---|---|---|---|---|---|---|---|---|---|---|---|---|---|
| RSTDR01-a | 13.40* | 0.26 Ma | 0.62 Ma | 64% | 0.046 | 0.009 | 0.4 | 87% | 7 | 23 | 13.39 | 0.37 Ma | 0.67 Ma | 293.3 | 3.5 | 52% | 0.5 | 77% | 14.57 | 0.29 Ma | 0.67 Ma | 0.021 |
| RSTDR01-c | Discordant | | | | | | | | | | | | | | | | | | | | | 0.076 |
| RSTDR01-d | 13.45* | 0.23 Ma | 0.61 Ma | 94% | 0.06 | 0.036 | 1.0 | 48% | 13 | 20 | 13.46 | 0.28 Ma | 0.63 Ma | 296.0 | 1.5 | 67% | 1.1 | 36% | 13.90 | 0.36 Ma | 0.68 Ma | 0.015 |
| RHRDV01-a | 7.46* | 0.41 Ma | 0.52 Ma | 58% | 0.032 | 0.005 | 1.0 | 45% | 7 | 23 | 7.46 | 0.47 Ma | 0.57 Ma | 294.3 | 1.3 | 77% | 1.2 | 31% | 6.53 | 0.58 Ma | 0.64 Ma | 0.018 |
| RHRDV01-b | 8.59 | 0.09 Ma | 0.37 Ma | 87% | 0.403 | 0.33 | 1.4 | 18% | 13 | 23 | 8.59 | 0.11 Ma | 0.38 Ma | 298.6 | 1.0 | 39% | 1.5 | 11% | 8.88 | 0.29 Ma | 0.47 Ma | 0.119 |
| RHRDV01-c | Discordant | | | | | | | | | | | | | | | | | | | | | 0.005 |
| RHRDV01-d | 0.42* | 0.23 Ma | 0.23 Ma | 78% | 0.151 | 0.105 | 0.9 | 56% | 9 | 23 | 0.41 | 0.23 Ma | 0.23 Ma | 293.5 | 1.1 | 1% | 1.3 | 27% | 0.98 | 0.37 Ma | 0.37 Ma | 0.023 |
| RJKBR01-a | 0.48 | 0.23 Ma | 0.23 Ma | 86% | 0.121 | 0.015 | 0.6 | 90% | 16 | 23 | 0.63 | 0.27 Ma | 0.27 Ma | 298.0 | 0.9 | 2% | 0.6 | 90% | | | | 0.107 |
| RJKBR01-c | Discordant | | | | | | | | | | | | | | | | | | | | | 0.096 |
| RJKBR01-g | 2.04* | 0.02 Ma | 0.09 Ma | 100% | 0.617 | 0.06 | 0.6 | 91% | 20 | 20 | 2.04 | 0.02 Ma | 0.09 Ma | 295.9 | 1.0 | 78% | 0.7 | 86% | 1.97 | 0.04 Ma | 0.09 Ma | 0.549 |
| RJKBR01-h | 1.79 | 0.76 Ma | 0.76 Ma | 84% | 0.096 | 0.079 | 1.6 | 8% | 14 | 23 | 0.23 | 0.13 Ma | 0.13 Ma | 300.3 | 4.0 | 0% | 1.8 | 5% | 3.66 | 0.69 Ma | 0.69 Ma | 0.019 |
| RJKBR01-k | 0.51* | 0.39 Ma | 0.39 Ma | 55% | 0.041 | 0.006 | 1.0 | 44% | 11 | 23 | 0.44 | 0.28 Ma | 0.28 Ma | 295.9 | 2.5 | 0% | 1.7 | 7% | 0.61 | 0.64 Ma | 0.64 Ma | 0.052 |
| RSTLK01-a | 10.68* | 0.16 Ma | 0.47 Ma | 98% | 0.064 | 0.021 | 1.4 | 17% | 15 | 19 | 10.68 | 0.20 Ma | 0.49 Ma | 295.6 | 2.0 | 62% | 1.5 | 9% | 10.98 | 0.24 Ma | 0.52 Ma | 0.038 |
| RPJRS01-a | 0.32* | 0.09 Ma | 0.09 Ma | 80% | 0.354 | 0.061 | 0.6 | 80% | 11 | 23 | 0.32 | 0.12 Ma | 0.12 Ma | 289.7 | 2.6 | 25% | 0.8 | 63% | 0.65 | 0.12 Ma | 0.12 Ma | 0.273 |
| RPJRS01-b | 0.64* | 0.17 Ma | 0.17 Ma | 98% | 0.052 | 0.013 | 1.1 | 37% | 12 | 17 | 0.64 | 0.18 Ma | 0.18 Ma | 295.4 | 0.8 | 5% | 1.3 | 22% | 1.79 | 0.43 Ma | 0.44 Ma | 0.041 |
| RPJRS01-c | 0.57* | 0.05 Ma | 0.06 Ma | 95% | 0.427 | 0.23 | 0.6 | 88% | 13 | 23 | 0.57 | 0.07 Ma | 0.07 Ma | 294.5 | 1.4 | 13% | 0.7 | 75% | 0.72 | 0.08 Ma | 0.08 Ma | 0.192 |
| RPJRS01-c | 0.44 | 0.10 Ma | 0.10 Ma | 100% | 0.073 | 0.056 | 1.0 | 45% | 23 | 23 | 0.45 | 0.12 Ma | 0.14 Ma | 298.4 | 3.2 | 10% | 1.1 | 34% | 0.62 | 0.18 Ma | 0.18 Ma | 0.035 |
| RPJRS01-e | 0.21 | 0.18 Ma | 0.18 Ma | 62% | 0.041 | 0.009 | 0.9 | 50% | 10 | 23 | 0.30 | 0.18 Ma | 0.18 Ma | 298.0 | 1.4 | 3% | 1.2 | 32% | | | | 0.052 |
| RPJRS01-f | Discordant | | | | | | | | | | | | | | | | | | | | | 0.109 |

**Table 2**: The $^{40}$Ar/$^{39}$Ar Incremental Heating Results for the Iceland Grains.
*Plateau calculated using the non-atmospheric intercept shown
i=internal uncertainty; f = full uncertainty
P=probability of fit factor
MSWD=Mean square of the weighted deviants
n=heating steps used in age calculations for both plateau and isochron
N=total heating steps analysed
SF=spreading factor
Negative total fusion ages are not shown

Two samples contained consistent plateaus with one intermediate temperature step containing a clear burst of excess $^{40}$Ar. This includes RSTDR01-a (Figure 3) and RHRDV01-b (Figure 4), both of which had plateau segments that contained >25% of $^{39}$Ar prior to and after the burst. Provided the nature of the samples (RSTDR01-a = holocrystalline basalt with olivine, plagioclase and clinopyroxene; RHDV01-b = glassy olivine basalt), and nature of the study (detrital geochronology) these single steps were excluded from the age calculation, but a plateau age is still provided using the apparent ages that precede and succeed the discordant steps. Here we define discordant as individual heating steps that statistically fall off the plateau.

Ten of the fifteen successful age determinations produced concordant sub-atmospheric $^{40}$Ar/$^{36}$Ar$_0$ (<298.56) inverse isochron intercepts (Table 2; Figure 8). The commonality of a sub-atmospheric intercept is consistent with the glassy nature of the samples, wherein slight kinetic fractionation of the lighter $^{36}$Ar occurs during the rapid lava quenching upon eruption (Dalrymple 1969; Renne et al., 2009). Sample plateaus that were recalculated using the measured sub-atmospheric intercept are shown with an * in Table 2 and Figures 3-7. No samples produced concordant supra-atmospheric intercepts, and the only potential evidence for trapped mantle derived argon were found in RSTDR01-a and RHRDV01-b. Except for the discordant RSTDR01-C (Figure 3), none of the analysed samples had heating spectrums that suggest significant $^{39}$Ar and $^{37}$Ar recoil patterns.

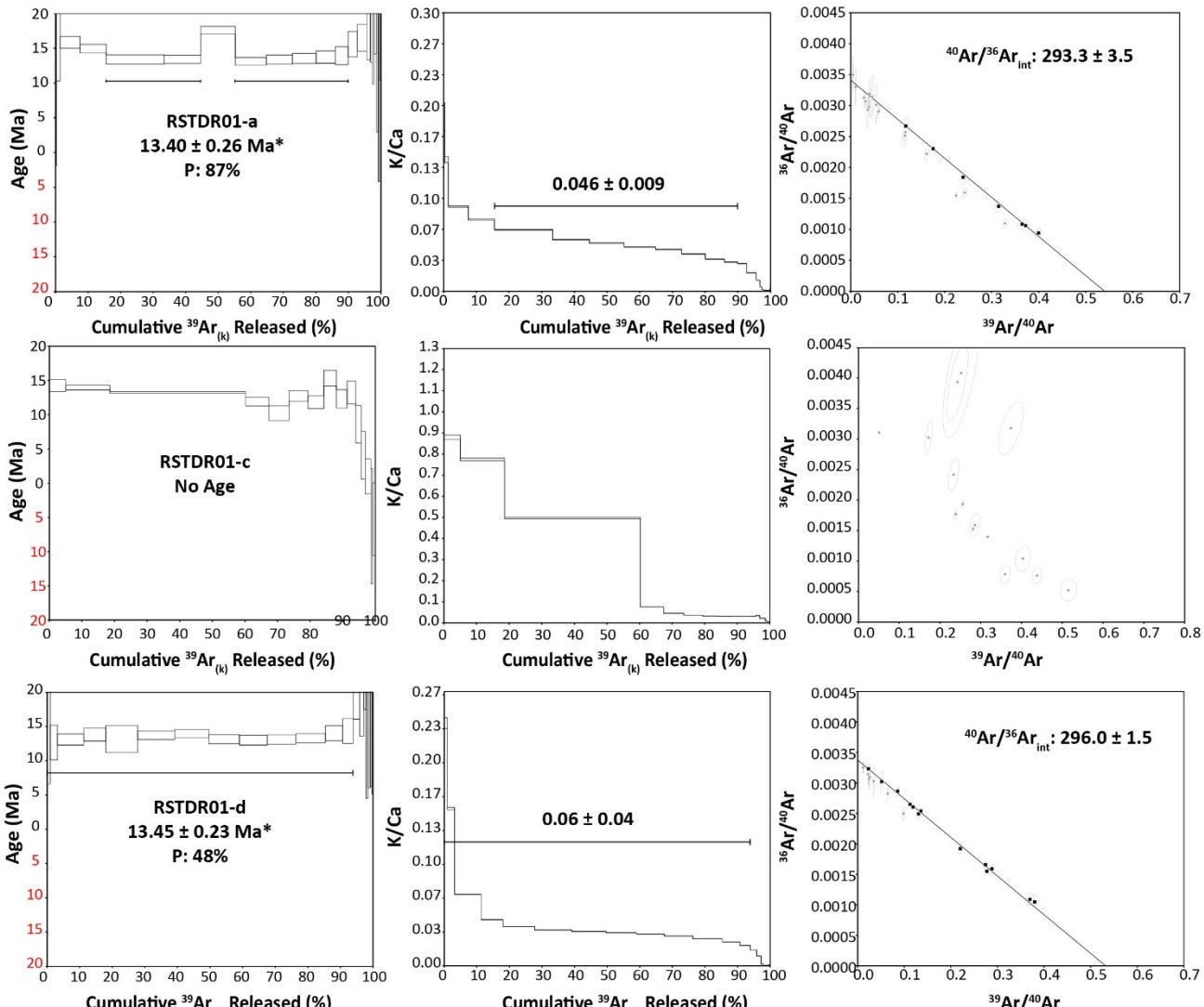

**Figure 3:** The $^{40}$Ar/$^{39}$Ar incremental heating experiment results for RSTDR01 grains. (Left) The heating plateau results. The line represents the steps used in the plateau age calculation. (Right) The inverse isochron results. The black squares represent steps used in the age determination calculation; grey squares are excluded steps. A * indicates the plateau age was recalculated using the measured $^{40}$Ar/$^{36}$Ar$_0$ shown. P is the probability of fit factor.

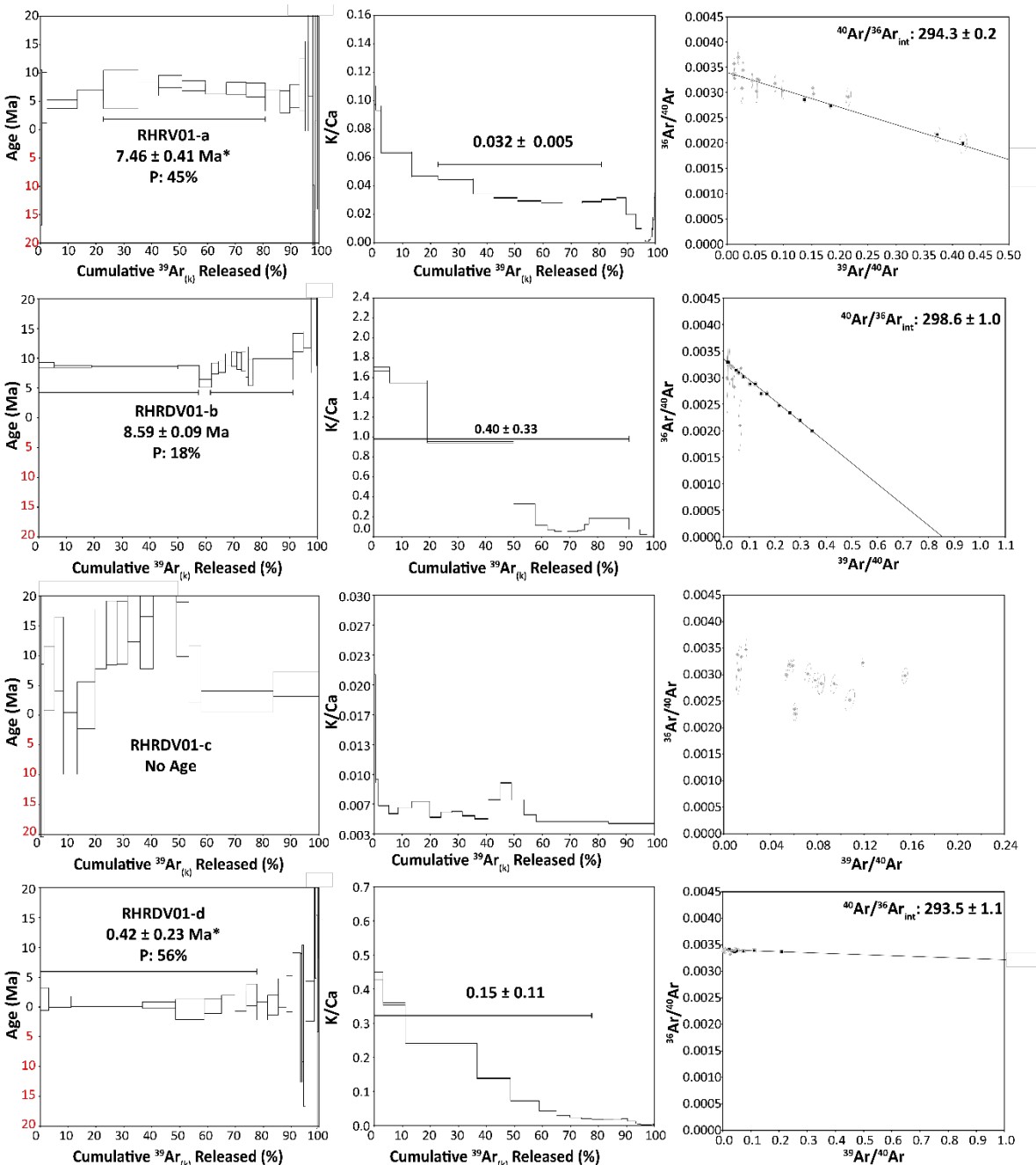

**Figure 4:** The $^{40}Ar/^{39}Ar$ incremental heating experiment results for RHRDV01 grains. (Left) The heating plateau results. The line represents the steps used in the plateau age calculation. (Right) The inverse isochron results. The black squares represent steps used in the age determination calculation; grey squares are excluded steps. A * indicates the plateau age was recalculated using the measured $^{40}Ar/^{36}Ar_0$ shown. P is the probability of fit factor.

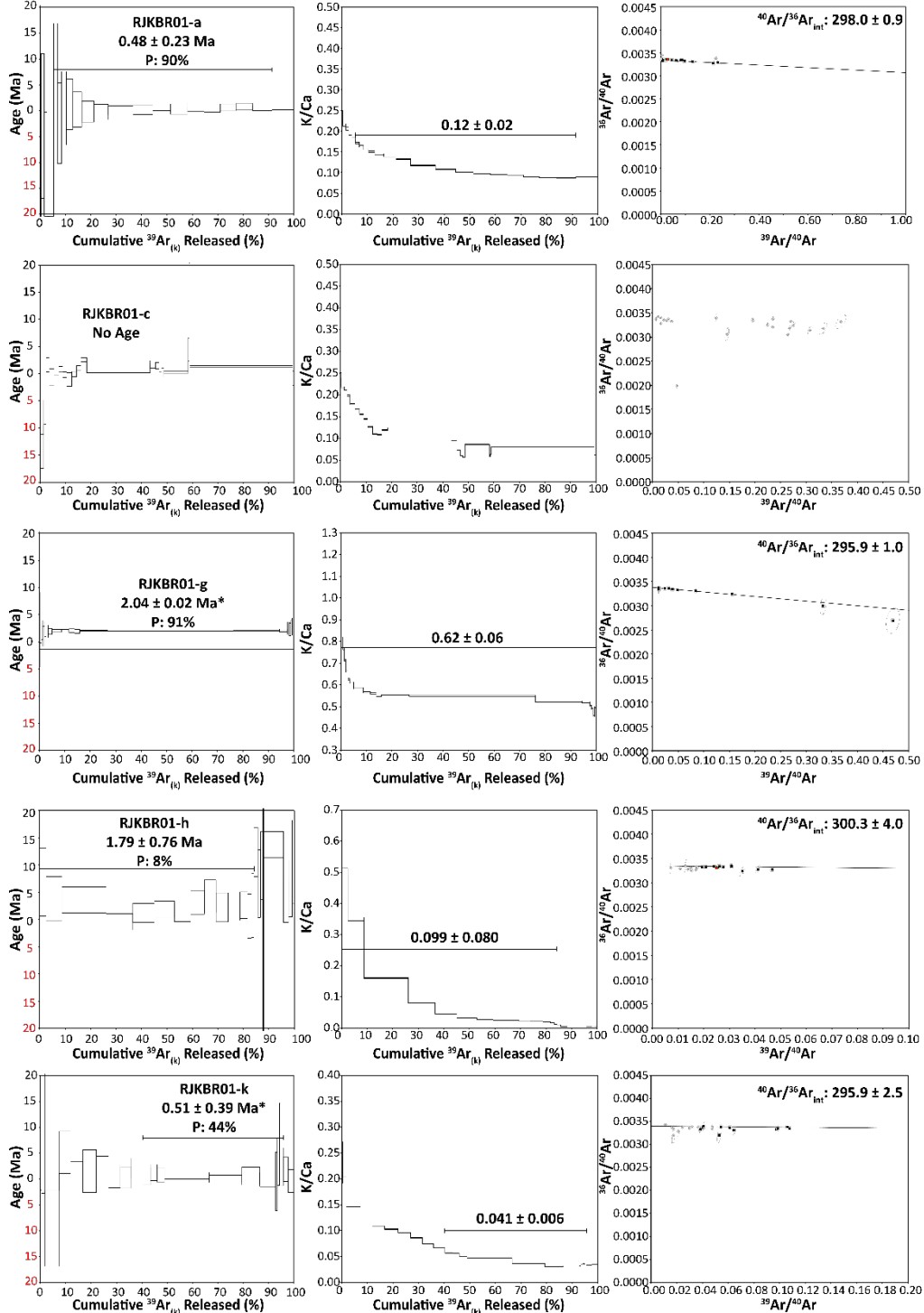

**Figure 5:** The $^{40}$Ar/$^{39}$Ar incremental heating experiment results for RKJBR01 grains. (Left) The heating plateau results. The line represents the steps used in the plateau age calculation. (Right) The inverse isochron results. The black squares represent steps used in the age determination calculation; grey squares are excluded steps. A * indicates the plateau age was recalculated using the measured $^{40}$Ar/$^{36}$Ar$_0$ shown. P is the probability of fit factor.

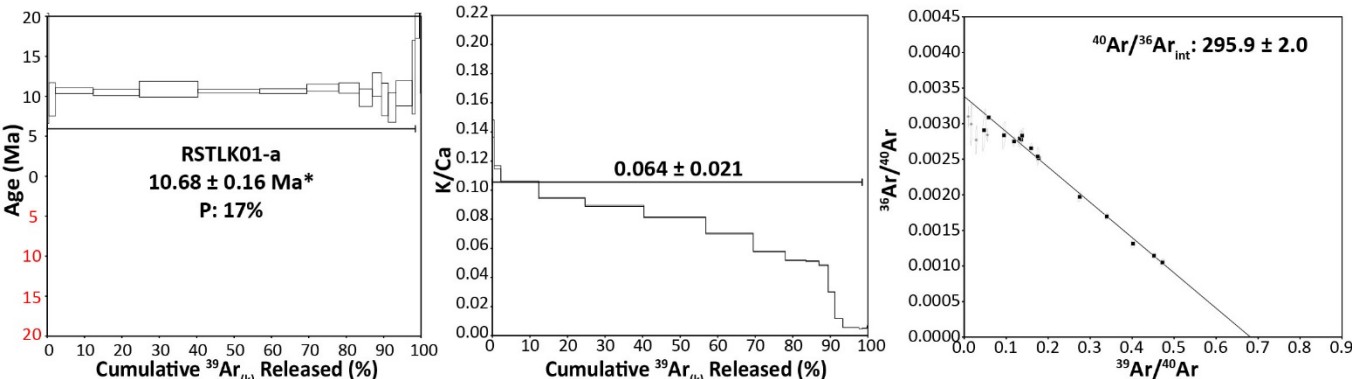

**Figure 6:** The $^{40}$Ar/$^{39}$Ar incremental heating experiment result for the single RSTLK01 grain. (Left) The heating plateau result. The line represents the steps used in the plateau age calculation. (Right) The inverse isochron result. The black squares represent steps used in the age determination calculation; grey squares are excluded steps. A * indicates the plateau age was recalculated using the measured $^{40}$Ar/$^{36}$Ar$_0$ shown. P is the probability of fit factor.

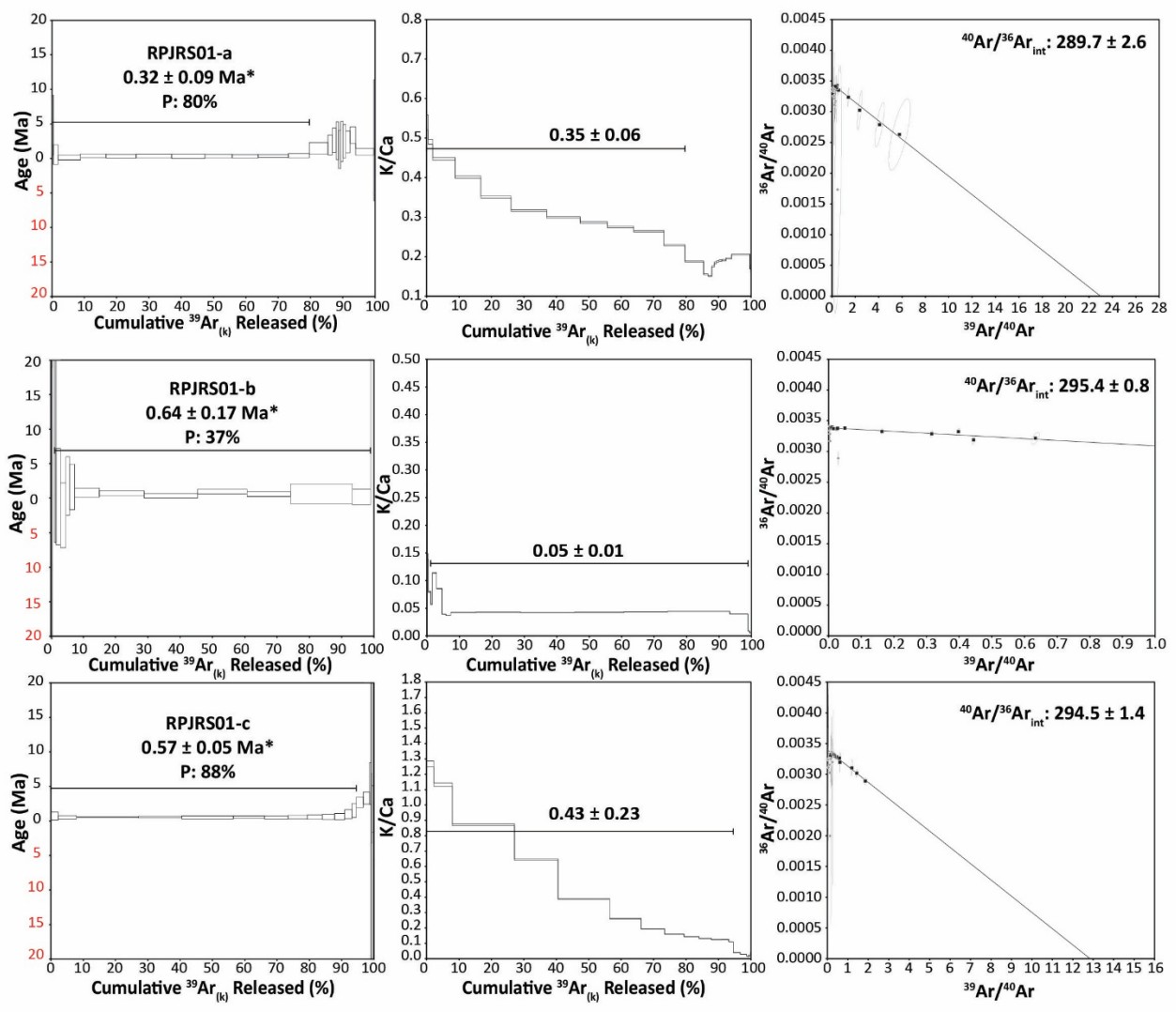

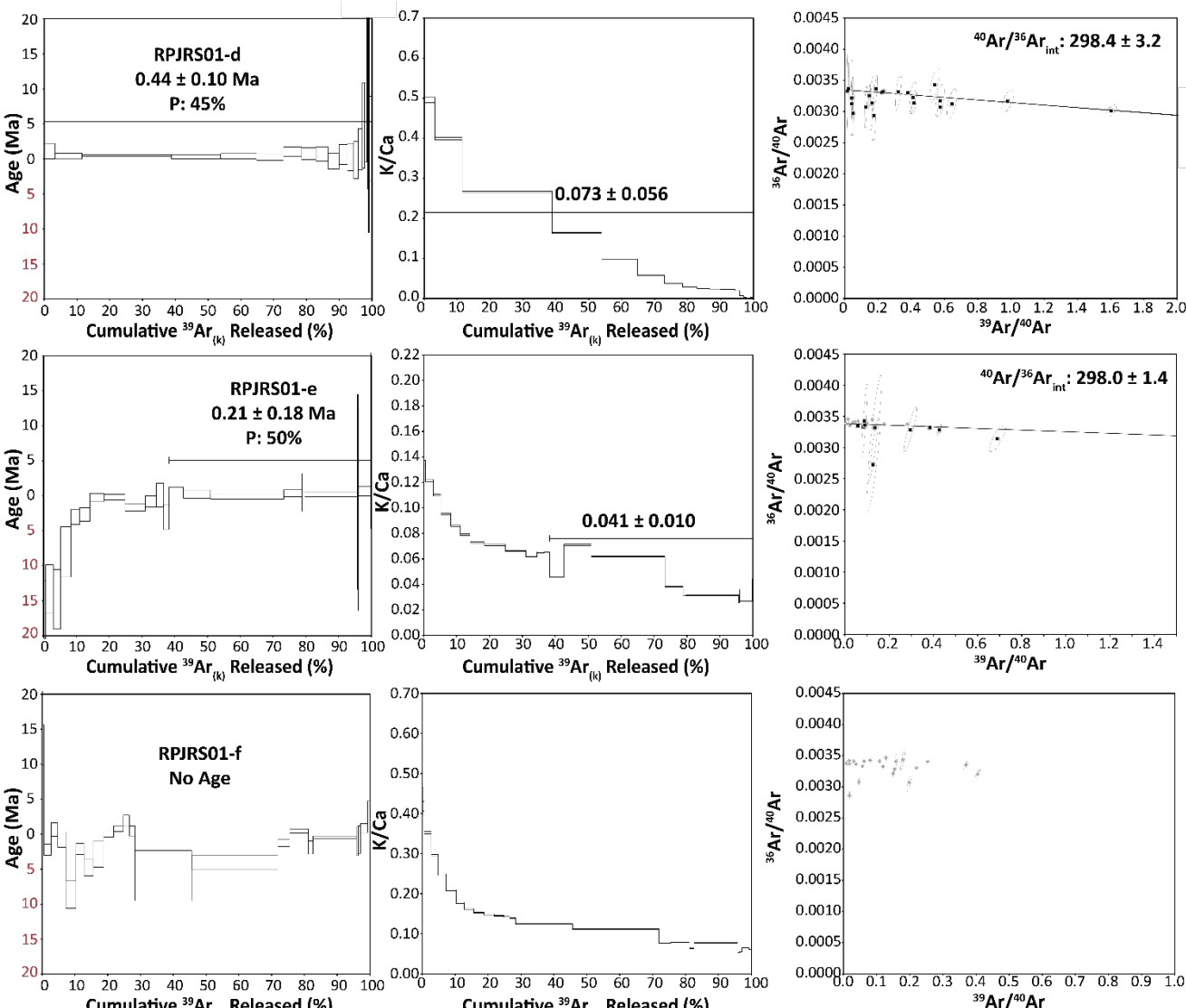

**Figure 7:** The $^{40}Ar/^{39}Ar$ incremental heating experiment results for RPJRS01 grains. (Left) The heating plateau results. The line represents the steps used in the plateau age calculation; grey squares are excluded steps. (Right) The inverse isochron results. The black squares represent steps used in the age determination calculation. A * indicates the plateau age was recalculated using the measured $^{40}Ar/^{36}Ar_0$ shown. P is the probability of fit factor.

## 4.0 Discussion

### 4.1 Individual Sediment Site Summaries

Here we discuss the similarity between DARL ages and expected volcanic bedrock ages within each sampled river watershed catchment area (Figure 1). It is important to note that the boundaries between each tectonic unit in Figure 1 are simplified and the contacts among the volcanic rock age units may be more diffuse than as displayed (Jóhannesson and Sæmundsson, 2009).

Out of the three grains that were analysed from the RSTDR01 sample (Stadará River), two produced concordant plateaus at *ca.* 13.4 Ma, and one produced a discordant plateau (Figure 3). The DARL ages are consistent with erosion of the middle Miocene volcanic bedrock terrain (10 to 15 Ma; Figure 1).

The RHRDV01 samples (Heradsvötn River) had three of four successful age determinations with two older (7.5 and 8.6 Ma) and one young (0.42 Ma) results (Figure 4). The older ages are not consistent with the currently proximal Pliocene-

upper Miocene volcanic bedrock terrain (3.3-5.5 Ma) or the younger terrain upstream. These unexpected results highlight a strength of the incremental heating DARL method. Having the ability to obtain more robust age spectrums than traditional K/Ar or total fusion methods allows for more detail questions to be asked on sediment transport mechanisms as opposed to simply discarding unexpected outliers assuming excess $^{40}$Ar. The younger DARL age could be derived from further upstream in the Mid-Iceland Belt region (<0.8 Ma). The discordant age spectrum and isochron from grain RHRDV01-c displayed a low

$^{40}$Ar*/$^{40}$Ar$_{atm}$ and K/Ca (0.005) and as such, small amounts of alteration and/or excess Ar could have greatly disturbed the apparent ages.

The RJKBR01 samples (Jökulsá á Brú River) had four out of five successful DARL age determinations, with ages ranging from 0.5 to 2.0 Ma (Figure 5). All the RJKBR01 ages are younger than the immediately proximal volcanic outcrops from the Pliocene-upper Miocene terrain (3.3-5.5 Ma) and upper Miocene (5.5-10 Ma) in the lower reaches of the watershed

(Figure 1). However, the Jökulsá á Brú River initiates near the glaciated Öræfi Volcanic Belt and runs through terrain believed to include < 3.3 Ma volcanic products (Figure 1). Therefore, the observed age ranges are consistent with the local river catchment geology. The analysed RJKBR01 grains were generally well behaved with one exception (Figure 5). RJKBR01-C had an erratic degassing pattern, wherein a very large concentration of atmospheric argon and $^{39}$Ar was released during the 1080°C, 1230°C and 1350°C heating steps, while other steps had low gas concentrations. The origin of this anomalous

degassing pattern is uncertain, but it may relate to pulsed disaggregation of the grain at certain temperatures, which results in a rapid release of trapped gas.

One grain was analysed from sample RSTLK01, producing a long concordant plateau with an age of 10.68 Ma (Figure 6). The measured DARL age is appropriate for the Störilækur River's catchment area that only includes 5.5 to 10 Ma volcanics (Figure 1).

Six grains were analysed from the Pjórsá River sample RPJRS01 with DARL ages that range from 0.2 to 0.6 Ma, with only one sample producing a discordant heating spectrum (Figure 7). The Pjórsá River runs between the Western and Eastern Volcanic zones and as such the young ages are consistent with erosion of Pleistocene volcanic terrains (Figure 1). The young ages of the sample produced large uncertainties (9 – 86%) with the largest uncertainty associated with the lowest K/Ca sample in the group (RPJRS01-e; 0.21 ± 0.18 Ma; K/Ca = 0.04). These samples also had low percentages of radiogenic $^{40}$Ar*

contributing to the higher uncertainty. The RPJRS01-e sample's low temperature steps correlate to a sub-atmospheric isochron and as such the difference between the $^{40}Ar/^{36}Ar_0$ released from the glassy mesostasis (sub-atmospheric) and the more crystalline mesostasis (atmospheric) resulted in a shorter plateau (62%). RPJRS01-f produced a discordant plateau that had negative apparent ages, likely due to having a sub-atmospheric intercept coupled with a very young age.

        In summary, the preliminary age data from this study indicates that the DARL method produces age determinations
consistent with what is expected for the river catchments based on available geological maps. The uncertainties on the ages are highly variable and typically quite high. However, this study is being carried out on mafic (primarily tholeiitic) terrain that ranges in age from Holocene to middle Miocene and as such represents a uniquely difficult region to perform DARL measurements. Below we discuss the technique further and provide recommendations for future improvements.

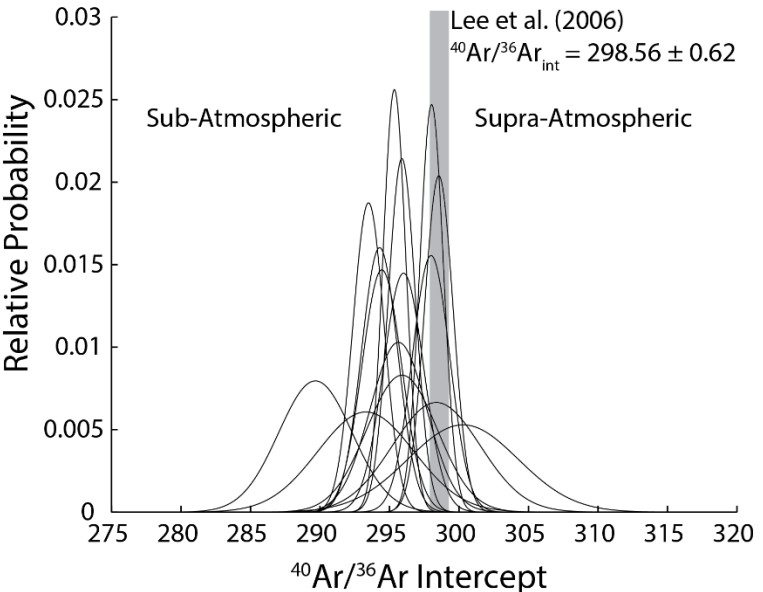

**Figure 8:** A probability distribution diagram of the calculated $^{40}Ar/^{36}Ar$ intercepts for the 15 successful age determination experiments. Note that many of the samples (typically glassy basalts) contain sub-atmospheric initial $^{40}Ar/^{36}Ar$ intercept values. The accepted modern $^{40}Ar/^{36}Ar$ of atmosphere (Lee et al., 2006) is shown with a grey bar.

**4.2 Additional Chemical Constraints**

Understanding the age distributions is valuable for constraining the changes in melt output rate and/or potential pulsation of regional volcanism. However, age alone does not speak to the changes in volcanic composition with time. Previous research on whole grain chemical analyses of Icelandic sediment provided valuable insights into regional compositional variation but could not be combined with age determinations (Thorpe et al., 2019). The $^{40}Ar/^{39}Ar$ method provides a means of assessing the ratio of K (through $^{39}Ar_K$ proxy) to Ca ($^{37}Ar_{Ca}$), which can provide a first order (e.g. low precision) assessment
of a variety of processes, including but not limited to, the degree of source melt enrichment, degree of mantle melting or the

assimilation/crystal fractionation history of the sample. Unfortunately, the grain K/Ca alone cannot differentiate between the possible origins. For example, Icelandic lava flows can have K/Ca ratios varying from near 0 to ~0.25 but fall on both the tholeiitic (aka sub alkaline) or alkaline evolution trends (Figure 9). Furthermore, the heating range employed in this study (400 – 1400°C), results in an underestimation of the clinopyroxene and olivine contributions to the total $^{37}$Ar released, indicating that the observed total gas K/Ca are likely an upper limit. However, K/Ca values would likely be useful in volcanic terrains that contain a range of igneous materials from mafic to felsic (e.g. continental arcs) or regions that display bimodal volcanic behavior (e.g. Snake River Plain).

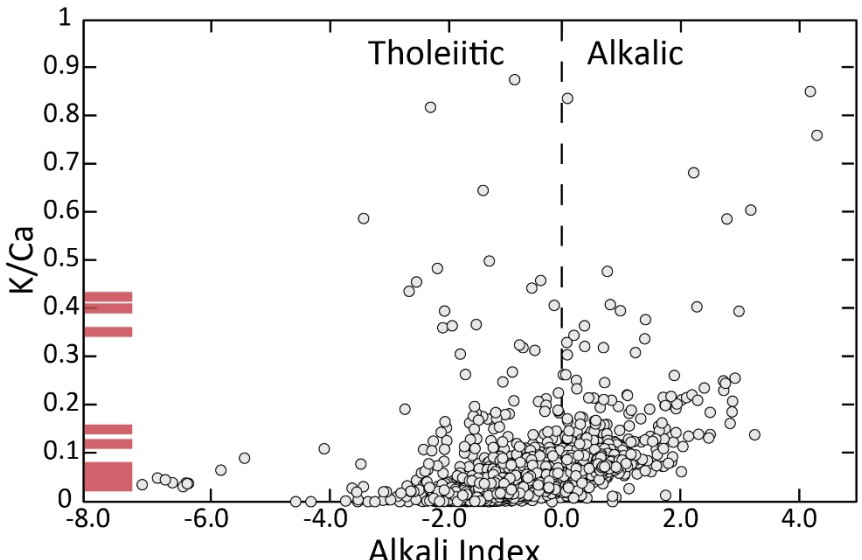

**Figure 9:** The bulk rock K/Ca and alkali index values for known Icelandic volcanics. The K/Ca values for the analysed grains are shown as red bars on the y-axis. Icelandic sample data from the compiled GEOROC database (DIGIS Team, 2023) and filtered to K/Ca <1 to remove rhyolite samples. The alkali index is calculated as [Na$_2$O + K$_2$O] – [SiO$_2$ x 0.369 - 14.350] (Rhodes et al., 2012).

Additional non-destructive (e.g. scanning electron microscopy [SEM] analyses), split grain (e.g. Ellis et al., 2017) or semi-destructive (e.g. laser ablation – inductively coupled plasma mass spectrometry [LA-ICPMS]) analyses prior to irradiation would be required to more thoroughly trace petrologic evolution. This would require lightly polishing one of the grain surface's and analysing the mineralogy using an SEM and/or performing spot analyses on glass using an electron microprobe or LA-ICPMS. Samples would then need to be carefully demounted and independently acid leached thoroughly to remove any epoxy or carbon coating prior to irradiation and $^{40}$Ar/$^{39}$Ar analyses. This coupled petrologic analysis and age determination on single grains would provide novel insights into the long-term first-order evolution of a volcanic terrain.

**4.3 Suggestions for More Rapid DARL Analyses**

A significant advantage of the incremental heating DARL method is the ability to filter out lithic grains that have been rendered discordant due to alteration and/or incorporation of trapped excess argon. However, this comes at a large cost to analysis time. A single incremental heating experiment using a vacuum furnace takes ~12 hours to complete. Therefore, a rapid analyses method is required to obtain the large N values needed for a successful detrital geochronology study. Trop et al. (2022) used incremental heating on a subset of grains to assess for alteration or excess argon. Thereafter, they employed the total fusion method wherein individual grains or multi-grain aliquots were fused in a single step (Trop et al., 2022). An atmospheric $^{40}Ar/^{36}Ar_0$ was assumed with the age calculations and the results were ~equivalent to K/Ar ages collected from the region. When the observed total gas age (calculated using an assumed $^{40}Ar/^{36}Ar_0 = 298.56 \pm 0.62$) for the new Iceland DARL ages are compared to the plateau age, an approximate 1:1 line is observed with a large statistical discordance (MSWD=32; Figure 10a). The < 1 Ma age determinations mostly fall off the ideal 1:1 line and commonly display negative total fusion ages. These negative ages are due to the combination of low $^{40}Ar^*$ and a sub-atmospheric $^{40}Ar/^{36}Ar_0$. When an atmospheric $^{40}Ar/^{36}Ar^0$ is assumed, the $^{40}Ar_{atm}$ contribution is overcorrected, resulting in negative $^{40}Ar^*$ values. These observations indicate that standard total fusion analyses would not be appropriate for terrains consisting of young and/or glassy lava flows.

| | Incremental Heating Results | | Total Fusion Results | | Partial Fusion (680 to 1140°C) | |
|---|---|---|---|---|---|---|
| | Age (Ma) | ± 2σ (i) | Age | ± 2σ (i) | Age | ± 2σ (i) |
| RSTDR01-a | 13.4 | 0.3 | 14.6 | 0.3 | 14.1 | 0.3 |
| RSTDR01-d | 13.5 | 0.2 | 13.9 | 0.4 | 13.8 | 0.6 |
| RHRDV01-a | 7.5 | 0.4 | 6.5 | 0.6 | 6.3 | 1.4 |
| RHRDV01-b | 8.6 | 0.1 | 8.9 | 0.3 | 11.2 | 2.2 |
| RHRDV01-d | 0.4 | 0.2 | -2.3 | 0.3 | 1.0 | 0.4 |
| RJKBR01-a | 0.5 | 0.2 | -0.7 | 0.5 | 1.8 | 1.7 |
| RJKBR01-g | 2.04 | 0.02 | 1.97 | 0.04 | 2.04 | 0.04 |
| RJKBR01-h | 1.8 | 0.8 | 3.7 | 0.7 | 5.2 | 1.5 |
| RJKBR01-k | 0.5 | 0.4 | 0.6 | 0.6 | 0.7 | 0.7 |
| RSTLK01-a | 10.7 | 0.2 | 11.0 | 0.2 | 10.6 | 0.3 |
| RPJRS01-a | 0.3 | 0.1 | 0.7 | 0.1 | 0.2 | 0.2 |
| RPJRS01-b | 0.6 | 0.2 | 1.8 | 0.4 | 0.4 | 0.8 |
| RPJRS01-c | 0.6 | 0.1 | 0.7 | 0.1 | 0.7 | 0.2 |
| RPJRS01-d | 0.4 | 0.1 | 0.6 | 0.2 | 0.8 | 0.3 |
| RPJRS01-e | 0.2 | 0.2 | -1.3 | 0.2 | 0.3 | 0.7 |

**Table 3:** A comparison of concordant plateau age, total fusion ages assuming all gas released and an atmospheric $^{40}Ar/^{36}Ar_0$ value and partial 'fusion' results for steps between 680° and 1140°C assuming a $^{40}Ar/^{36}Ar_0$ of 296 ± 4. Samples with discordant heating spectrum are excluded from the table.

An alternative method, that combines aspects of the incremental heating and total fusion DARL methodologies (e.g. Trop et al., 2022), could provide a more robust detrital geochronology history. This prospective method would first involve

analysing a representative 5-10 grains from a site using the incremental heating method. This provides a first order assessment of the degree of alteration and prevalence of non-atmospheric $^{40}Ar/^{36}Ar_0$ intercepts for the site. Based on that knowledge, an ideal temperature range and an assumed $^{40}Ar/^{36}Ar_0$ with a larger uncertainty can be employed to calculate the partial-fusion ages. The bulk of the Icelandic grains contained plateaus that incorporated heating steps in the 680 to 1140°C range (Figures 3 – 7). Therefore, we can calculate the partial-fusion age between those temperature steps, using an $^{40}Ar/^{36}Ar_0$ that is representative of our dataset (296 ± 4; Figure 8). Figure 10b shows the correlation between the plateaus and the equivalent total fusion values (calculated using total gas values from the 680-1140°C steps). It is important to note that the partial fusion errors here are overestimated due to expected lower blank corrections (using a single preceding blank instead of a polynomial fit to multiple blanks) and peak regression uncertainties (higher peak signals released in a single heating step) during an actual fusion measurement. This method appears to provide a much better concordance between a modelled partial-fusion measurement and the detailed plateau age with a MSWD of 3.7. This method also provided much better fits for the young glassy basalt samples; however, the large uncertainties indicate the method can only be used to interpret regional changes on the Ma timescale. Based on these preliminary Icelandic results, our next recommended steps would be to pre-heat the grains to 680°C while under active vacuum, then perform a single 1140°C heating step to obtain the age of the grain. This will allow for 100s of grains to be analysed within a reasonable timeframe — providing the large N values needed for a detrital geochronology experiment. Alternatively, since the sensitivity to the sub atmospheric intercepts seems greater in the youngest samples, perhaps the alternate $^{40}Ar/^{36}Ar_0$ (296 ± 4) should only be used when a sample produces a negative age result. Provided that most outcrops in Iceland contain low K mafic lava flows, this method should be more precise in other volcanic settings such as continental arcs (e.g. Trop et al., 2022) or among more alkalic ocean islands (e.g. Samoa).

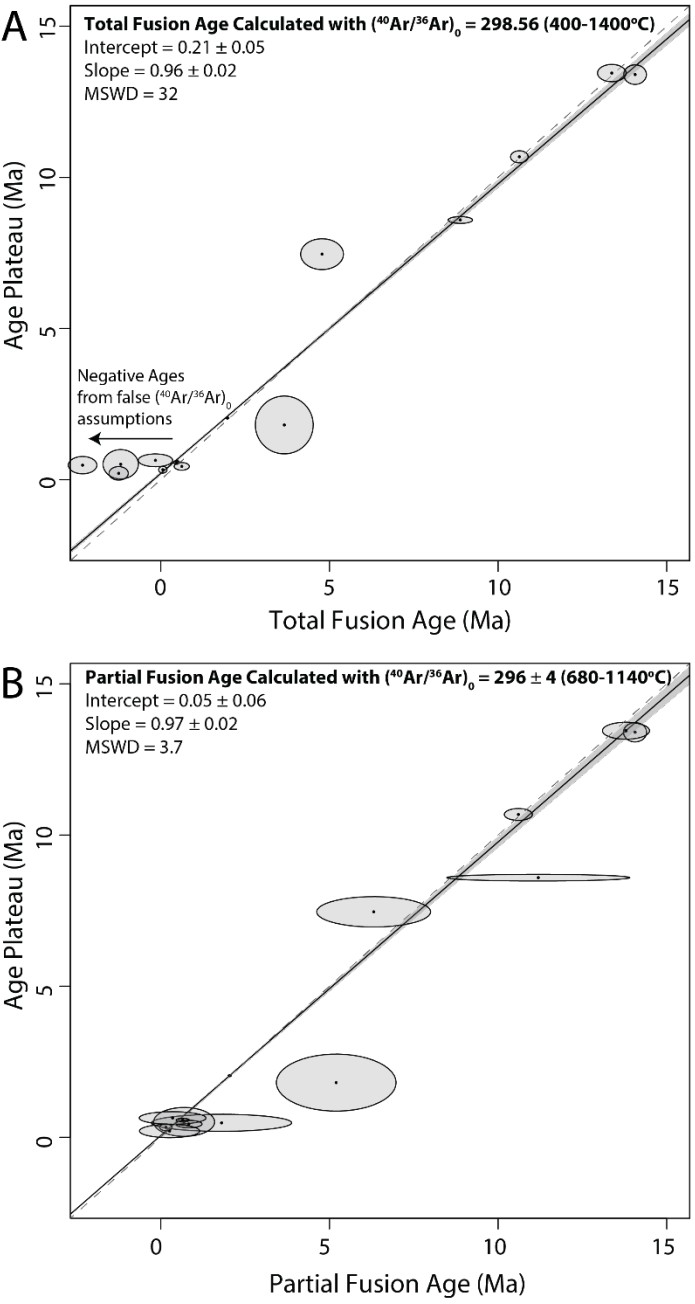

**Figure 10:** A comparison of apparent total fusion ages (calculated using the summed gas of all temperature steps) versus the plateau age for the Icelandic grains. (A) A comparison of fusion ages to plateau with an assumed $^{40}Ar/^{36}Ar_0$ of $298.56 \pm 0.62$ (Lee et al., 2006). (B) The partial fusion (680-1140°C) ages compared to the plateau age. The partial fusion ages are calculated using an assumed $^{40}Ar/^{36}Ar_0$ of $296 \pm 4$. The best fit line slope and intercept are calculated using a York style linear regression (York, 1968). A reference 1:1 dashed line is included for comparison.

320

## 5.0 Conclusions

The DARL method (Trop et al., 2022) provides a novel means of constraining the volcanic history of a region through detrital geochronology of lithic grain sand/fine gravel samples. The method is particularly applicable to regions that contain mafic and fine-grained extrusive lithologies, which are often underrepresented using traditional detrital geochronology minerals such as zircon, apatite, sanidine and hornblende. Here we show that modern high-sensitivity mass spectrometers, such as the NGX, allow for incremental heating experiments to be carried out on young whole rock volcanic grains. New results from fifteen of nineteen lithic grains from Icelandic River sands provided statistically reliable age determinations that reflect the erosion of Pleistocene to Miocene basalt dominated catchment areas. Although the internal concordance test afforded by the incremental heating method has many advantages, the long analyses time hinders the method's use for detrital geochronology studies, which rely on high N values. Therefore, we propose that a subset of grains from a sampling site be analysed with the incremental heating method in order to define the best partial fusion temperature ranges and appropriate assumed $^{40}Ar/^{36}Ar_0$. More work is required to assess the validity of the method in different geologic settings, but the primary data from this study indicates the method is valid and allows for detrital geochronology studies of dominantly mafic terrains.

## Author Contributions

O. Okwueze prepared and analysed the samples, regressed the data and wrote the bulk of the manuscript. K. Konrad designed the experiment, supervised the analyses and contributed to the discussion. T. Capaldi collected the samples and contributed to the discussion.

## Financial Support

This material is based upon work supported by the National Science Foundation under Grant Nos. EAR-1759200 and EAR-1759353. Any opinions, findings, and conclusions or recommendations expressed in this material are those of the authors and do not necessarily reflect the views of the National Science Foundation.

## Competing Interest

The authors declare that they have no conflict of interest.

## Code and Data Availability

All new data is provided in Tables 1, 2 and 3 with detailed age results provided in the supplementary materials document.

## Acknowledgements

Thanks to the AGeS program for its support of K. Konrad and O. Okwueze. This work represents the cumulation of undergraduate research performed by O. Okwueze. Kathy Zanetti is thanked for assistance with the age determinations. Margo Odlum, Barbara Kleine-Marshall, and Edward Marshall are thanked for field assistance and logistics. This work greatly benefited from reviews and suggestions from Drs. Matt Brueseke, Jeff Benowitz, Jeffrey Trop and Dan Barfod.

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
