# Peer review of "New Developments in Incremental Heating Detrital 40Ar/39Ar Lithic (DARL) Geochronology using Icelandic River Sand"

_EGUsphere, 2024_

## Referee Comment (RC1)

We (Brueseke, Benowitz, Trop) found, the currently under discussion manuscript, *New Developments in Incremental Heating Detrital $^{40}Ar/^{39}Ar$ Lithic (DARL) Geochronology using Icelandic River Sand by* Odinaka Okwueze, Kevin Konrad, and Tomas Capaldi well written and a good contribution to the continued use of the DARL (Detrital Argon Lithics) geochronology approach. We agree the magmatic history of the glaciated Iceland magmatic province will benefit from applications of the DARL technique, as will other relatively remote and glaciated area such as the Cascades Arc of Northwestern United States.

We graciously recommend some key adjustments to the text, given our and others past work doing both $^{40}Ar/^{39}Ar$ step-heating and modified single grain fusion on ground mass, whole rock chips, and discrete mineral grains from gravel- and sand-sized volcanic-lithic clasts.

We first reported $^{40}Ar/^{39}Ar$ ages on volcanic-lithic grains from modern river sands in the Wrangell Volcanic arc at a 2014 conference (Benowitz et al., 2014), where we demonstrated that a modified heating schedule of sand-sized volcanic lithics was more efficient and accurate for DARL analyses. This was based on incremental heating single sand-sized volcanic-lithic grains and then modifying our fusion schedule based on these results. We also recommended when applying DARL to other regions standard step-heating be performed before developing a fusion or modified (shortened) step-heat schedule. At the time we were concerned about excess $^{40}Ar$ not excess $^{36}Ar$ (which Okwueze et al. document). We agree that excess $^{36}Ar$ is an underappreciated aspect of $^{40}Ar/^{39}Ar$ geochronology (Benowitz et al., 2018). These method details were explained in a subsequent Geosphere article (Trop et al., 2022; relevant aspects are copied below) and inasmuch, should be noted as where the DARL technique originated and was first published. Furthermore, Kenny et al. (2022) also performed $^{40}Ar/^{39}Ar$ incremental step-heats on detrital sand volcanic-lithic grains. We also performed and published (Trop et al., 2022) incremental step-heats on volcanic-lithic grains when results were questionable or were of key age spans as one of our goals was to determine the age of initiation for the Wrangell Arc. VanderLeest et al (2020) also applied step-heats to detrital clasts.

Thus, we kindly suggest that Okwueze et al. revise their text and clarify that $^{40}Ar/^{39}Ar$ step-heats and modified fusions were done previously on modern river volcanic-lithic grains, consequently the contribution here builds on these prior studies. This key fact should be made clearer in this manuscript; as-is, the DARL technique as described is not new or particularly novel, especially given that it is centered on n = 15 grains (vs. n = ~2600 grains; Trop et al., 2022). Additionally, Kenny et al., 2022 performed modified step-heats on 50 grains with step counts varying from 2 to 15 (?) steps to optimize number of grains vs. diffusion profile information. See their supplemental files.

We understand there is so much literature out there, that it is easy to miss aspects of past research and take no offense and based on conversations with the corresponding author know none was meant. We are genuinely excited to see more DARL work reported from this research team and others.

**Specific recommended changes:**

Something like the following for their introduction: *Following previous combination $^{40}Ar/^{39}Ar$ incremental step-heating and informed modified fusion procedure on modern river volcanic-lithic grains (Benowitz et al., 2014, Trop et al., 2022), we developed a new DARL partial fusion procedure specific to the magmatic products of Iceland.*

Below are additional changes and information re: relevant past work we recommend the authors consider during their revision.

**Around line 15** (Benowitz et al., 2014; VanderLeest et al., 2020; Kenny et al., 2022; Trop et al., 2022 did $^{40}Ar/^{39}Ar$ incremental step-heats on detrital cobbles and/or sand). Here we present a new methodology for capturing the magmatic history of fine grained extrusive volcanic rocks using single grain detrital $^{40}Ar/^{39}Ar$ incremental heating geochronology. The DARL (or Detrital Argon Lithics) **method thus far** has consisted of $^{40}Ar/^{39}Ar$ total fusion analyses, which pose a problem in the case of Iceland, due to the nature of its young glassy lava flows commonly displaying subatmospheric $^{40}Ar/^{36}Ar$ isochron intercepts and low $^{40}Ar*$.

**Around line 25** Benowitz et al., 2014**;** Trop et al., 2022 did both a combination of informed single grain fusions based on incremental heating results; n = ~2600 grains are what was eventually analyzed and reported in Trop et al (2022)

For this reason, **we propose combining the aspects of the total fusion and incremental heating** DARL methodologies to acquire age data for the large N values needed for detrital studies while improving the accuracy of total fusion DARL analysis.

**Around line 40 (**DARL has been applied to sand and pebble grains and cobbles, and as a combination of modified fusion and incremental step-heating…. Benowitz et al., 2014; VanderLeest et al 2020; Kenny et al., 2022; Trop et al., 2022)

The detrital $^{40}Ar/^{39}Ar$ lithic (DARL) method is a relatively new detrital geochronological tool that **thus far** employed $^{40}Ar/^{39}Ar$ total fusion analyses on single grains or multi-grain aliquots recovered from cobble sized (>10 cm) volcanic sediments (Trop et al., 2022; Brueseke et al., 2023).

**Around line 50 (**this has already been done…Benowitz et al., 2014; Kenny et al., 2022; Trop et al., 2022**)**

Here **we expand** upon the method through incremental heating experiments on single coarse sand grains of volcanic lithic fragments from Icelandic rivers.

**Line 220 (**This seems a little overstated given ~10 years of DARL step-heating work and the orders of magnitude larger number of individual DARL analyses from Trop et al., 2022 and the combination of geochemistry and DARL dating in VanderLeest et al 2020 and Brueseke et al., 2023).

Provided the level of difficulty, the incremental heating DARL experiments worked well and **represent an advancement in the field of detrital geochronology.**

**Around line 250** (this was sort of done -Trop et al., 2022- to evaluate alteration and excess $^{40}$Ar and for sure the DARL method has been applied to dominantly mafic bedrock sources.)

Although the internal concordance test afforded by the incremental heating method has many advantages, the long analyses time hinders the method's use for detrital geochronology studies, which rely on high= N values. Therefore, **we propose that a subset of grains from a sampling site be analyzed with the incremental heating method** in order to define the best partial fusion temperature ranges and appropriate assumed $^{40}$Ar/$^{36}$Ar0. More work is required to assess the validity of the method in different geologic settings, but the primary data from this study indicates the method is valid and **allows for detrital geochronology studies of dominantly mafic bedrock sources.**

**Around Line 260** (at the time we used 295.5 for atmospheric $^{40}$Ar/$^{36}$Ar0)…which now is not standard…but does make the reference to our work a bit confusing…perhaps remove?).

An atmospheric $^{40}$Ar/$^{36}$Ar0 was assumed with the age calculations and the results were ~equivalent to K/Ar ages collected from the region.

**Around line 295**

The DARL method provides a novel means of constraining the volcanic history of a region through detrital geochronology of lithic grain sand samples.

Please Add the reference to Trop et al. (2022), given that is where the DARL technique originated and was first published:

The DARL method (***Trop et al., 2022***) provides a novel means of constraining the volcanic history of a region through detrital geochronology of lithic grain sand samples.

**Other manuscript notes that need to be addressed:**

Please define what you mean by discordant: We think we know what you are referring, but it is never defined/explained how you are applying this broad term.

Table 1: Please add the known age range for magmatism for each sample/drainage.

How often did you measure mass discrimination? Did it drift? Could applying the "incorrect" mass discrimination explain your excess $^{36}$Ar (and excess $^{40}$Ar) measurements?

$^{36}$Ar was measured on a more sensitive electron multiplier? Where $^{40}$Ar was measured on a sensitive (but less so?) faraday? Is this a factor in the excess $^{36}$Ar measurements?

We doubt these are controlling factors on the excess $^{36}$Ar measurements…but these factors should be at least documented and mentioned-dismissed in the text/methods.

Perhaps more discussion on how modern mass spectrometer instrumentation allows for the clearer identification of excess $^{36}$Ar could be added?

**Around Line 275**

Therefore, we can calculate the partial-fusion age between those temperature steps, using an $^{40}$Ar/$^{36}$Ar$_0$ that is representative of our dataset (296 ± 4; Figure 8).

What was the range of determined $^{40}$Ar/$^{36}$Ar0 for all the grains analyzed?

289.7 to 300.3….Is it really sensible to assume a single subatmospheric $^{40}$Ar/$^{36}$Ar$_0$ for all samples? Given most results approximated or were greater than 298.56 ± 0.62 (Lee et al., 2006)?

The 296 ± 4: Is that a weighted average? The uncertainty is propagated during the age calculations?

**Isochron plots:**

Are the same steps used for the plateau age determinations used for the isochron age determinations? They should be. It seems for some of the samples this is not the case? It is hard to tell given the number of steps used in the isochron determinations are not listed in table 2. If always the same number of steps/same steps are used for isochron regressions as were used for the plateau age determinations (as they should be? Unless justified), please mention in text.

**RPJSO1-e**

Would you consider this stepping up age spectrum indicative of loss? If so, is it appropriate to perform a regression back to initial $^{40}$Ar/$^{36}$Ar (Isochron plot) given the documented loss?

**Figure 10**

This is a key figure…but we don't see the negative original age determinations in Table 2 and there are no supplemental isotopic files. Please add the negative (original) age determinations to Table 2 and add full supplemental files. Schaen et al. (2021) community based (dozens of noble gas lab authors) makes a strong case and sets out examples of how $^{40}$Ar/$^{39}$Ar isotopic information should be documented in scientific manuscripts. Regardless if Schaen et al. (2021) is followed to the "T", detailed isotopic tables are required to be included with $^{40}$Ar/$^{39}$Ar geochronology publications to be able to evaluate the authors results/interpretations/methods.

We see on table 2 you correct for excess $^{36}$Ar, but don't correct for excess $^{40}$Ar. Would it be better to use the original isochron age determinations for all analysis instead of plateau ages?

Figure 10….B looks stretched? i.e., Why are uncertainties so big?

Or are uncertainties blown up with the applied $296 \pm 4$ $^{40}Ar/^{36}Ar_0$, hence MSWD goes down simply because of the larger uncertainties? Compared to $298.56 \pm 0.62$ $^{40}Ar/^{36}Ar_0$ (Lee et al., 2006).

What would be the MSWD for graph A be if the youngest three ages were parsed? Seems those are biasing everything and for graph B all the ages are being modified (some far away from there "actual ages"!!!).

**Can you please add a table** of original ages/uncertainties for all samples vs. modified ages/uncertainty with the assumed $296 \pm 4$ $^{40}Ar/^{36}Ar_0$ determination. We think this is a key aspect…Yes you are shifting the youngest ages, but you are also shifting the other ages, Is that appropriate given the large variations in actual measured/calculated $^{40}Ar/^{36}Ar_0$?

For example, on figure 10….sample RHDRV01-b gets shifted from a total fusion age of 8.9 Ma ($\pm 0.03$) with a small uncertainty and becomes >11.0 Ma on 10B with a huge uncertainty ($\pm \sim 5$ Ma?).

Is this an improvement over the original accurate and precise age determinations?

Can you get negative ages simply due to statistics? i.e. An age result of 10 ka $\pm$ 20 ka on a lava means given enough analyses you would get a negative age from the same sample.

We are not sure if trying to make "exact" geologic interpretations from modified negative $^{40}Ar/^{39}Ar$ follows best practices. Yes these grains are young and the authors can robustly state that, but we are not sure applying a $296 \pm 4$ $^{40}Ar/^{36}Ar_0$ to a negative age with a measured 289.67 $^{40}Ar/^{36}Ar_0$ makes for a geologically more meaningful age.

Rough figure showing large shift from measured to modeled ages.

[Figure]

Is 1.79 Ma age on Figure 10? (RJKBR01-h)? might be…. forgive us if it is.

**Line 80** (>2 mm sized grains are granule sized gravel as opposed to sand sized grains, so the text should state that fine gravel (or granules) and sand was analyzed**).**

The bulk sediment samples were sieved and grains from the **2-3 mm size fraction** were selected for all sites except RJKBR01, where the 1-2 mm size fraction was used. Each selected grain was separated and given a unique identifier (i.e. -A; Figure 2).

**Data Availability**

Please include a link to all isotopic information (preferably in excel format) and supplemental figures using a file-sharing site like https://zenodo.org/records/802100.

As is, it is impossible to replot the presented data, evaluate the results, etc.

**Summary Suggestion:**

Perhaps a better DARL method for Iceland would be to: Degass/not measure/pump out lower temperature steps (below 680 °C). And then a apply a $296 \pm 4$ $^{40}Ar/^{36}Ar_0$ for the negative age determinations: but acknowledge these modeled age determinations are approximations and not indicative of exact geological eruptive events.

**Review References:**

Benowitz, J.A., Davis, K.N., Brueseke, M.E., Trop, J.M., and Layer, P., 2014, Investigating the lost arc: Geological constraints on ~25 Million years of magmatism along an arc-transform junction, Wrangell Volcanic Belt, Alaska, Geological Society of America Abstracts with Programs, Vol. 46, No.6, p.363.

Benowitz, J.A., Miggins, D.P., Koppers, A.A. and Layer, P.W., 2018, November. Why are some young volcanic rocks undateable: Chemistry, Environment, or instrumentation?. In *GSA Annual Meeting in Indianapolis, Indiana, USA-2018*. GSA.

Kenny GG, Hyde WR, Storey M, Garde AA, Whitehouse MJ, Beck P, Johansson L, Søndergaard AS, Bjørk AA, MacGregor JA, Khan SA. 2022, A Late Paleocene age for Greenland's Hiawatha impact structure. Science Advances.

Schaen, A.J., Jicha, B.R., Hodges, K.V., Vermeesch, P., Stelten, M.E., Mercer, C.M., Phillips, D., Rivera, T.A., Jourdan, F., Matchan, E.L. and Hemming, S.R., 2021. Interpreting and reporting 40Ar/39Ar geochronologic data. *GSA Bulletin*, *133*(3-4), pp.461-487

Trop, J.M., Benowitz, J.A., Kirby, C.S. and Brueseke, M.E., 2022. Geochronology of the Wrangell Arc: Spatial-temporal evolution of slab-edge magmatism along a flat-slab, subduction-transform transition, Alaska-Yukon. *Geosphere*, *18*(1), pp.19-48.

VanderLeest, R.A., Fosdick, J.C., Leonard, J.S. and Morgan, L.E., 2020. Detrital record of the late Oligocene–early Miocene mafic volcanic arc in the southern Patagonian Andes (~ 51° S) from single-clast geochronology and trace element geochemistry. *Journal of Geodynamics*, *138*, p.101751.

**See Trop et al 2022 clipped below**

This study reports a total of 3940 sand-sized DZ U-Pb, 2640 sand-sized DARL $^{40}$Ar/$^{39}$Ar, and 131 cobble-sized DARL $^{40}$Ar/$^{39}$Ar dates from modern sediment from 22 major rivers and eight tributaries. Figure 4 summarizes the geology with the watersheds that were sampled. Figures 5–9 display relative age probability plots of modern river sediment samples. Figures 10–12 display composite probability plots of all samples. Figures 13–15 show the spatial distribution of <35 Ma detrital dates. The following sections summarize key age results from the overall study region followed by age patterns from five sub-regions.

**We developed a procedure to limit the effects of alteration by degassing each sample at 0.5 W for 60 seconds,** and the released gas was pumped off for time efficiency and hence increased throughput. The results have a single-grain and/or multi-grain precision of 1%. **A subset of 14 samples was selected for higher-precision ages and step-heated** from relatively low temperatures until reaching fusion temperatures using the 6 W argon-ion laser (Benowitz et al., 2014). For each step, isotopic ratios of Ar were determined, with a range of mean square of weighted deviates (MSWD) values of 0.0 –6.25 (Table S1).

**See: Supplemental S6 and S9 from Trop et al., 2022**

The majority of samples were analyzed as single-grain or multi-grain fusion analysis approach. We developed a procedure to limit the effects of alteration by degassing each sample at 0.5 watts for 60 seconds, and the released gas was not measured and pumped off for time efficiency and increased throughput. The results have a single-grain and/or multi-grain precision of 1%. Two different batches were dated from the Cheslina River sand sample (and results were combined); 1000 to 1200 micron sized grains yielded better analytical returns than 500 to 1000 micron sized grains owing to the dominance of young (<1 Ma) volcanic bedrock with limited radiogenic 40Ar in the watershed. **Samples selected for further geochronology analysis were step-heated from relatively low temperatures until reaching fusion temperatures** using the 6-watt argon-ion laser (Śliwiński et al., 2012; Benowitz et al., 2019). Refer to Repository Items DR7–11 for full 40Ar/39Ar analytical results.

From Benowitz GSA 2014 Poster: Where we step-heated sand grains before developing a specific DARL technique for the Wrangell Arc.

[Figure]

**Figure 8:** We developed a modified $^{40}Ar/^{39}Ar$ single grain fusion approach to analyze detrital volcanic lithic grains. Representative $^{40}Ar/^{39}Ar$ age spectra from step-heated detrital volcanic-lithic grains from the Chisana River, which drains the north flank of the Wrangell volcanic field (Fig. 6). Based on these results, before doing single grain fusions of lithic grains we degas each grain at 500 mW without measuring the gas. This limits the affects of alteration, doesn't affect the age of unaltered grains, and is time efficient.

Chisana River $^{40}Ar/^{39}Ar$ WR

---

## Referee Comment (RC2)

Review : New Developments in Incremental Heating Detrital 40Ar/39Ar Lithic (DARL) Geochronology using Icelandic River Sand, Okwueze et al. (2024).

Overview :

This contribution attempts to improve on the DARL method by overcoming the inherent limitations in previous applications that employed exclusively K-Ar methods. This is done using a detailed 40Ar/39Ar step heating approach.

Much of the data are of high quality and show excellent release spectra. Data from some of the more complex step heating spectra are reasonably discussed.

Application of non-atmospheric trapped compositions for correcting plateau ages is also explained and justified. However, in the case of apparent non-atmospheric trapped components, I would use the isochron ages as these will be less affected by the trapped component issues.

The proposed method of 'partial fusion+averaged trapped component' is very poorly explained and it's implications are not clear. As read in the text, it appears to give a different, but similarly blurred picture of the age distribution of clastic materials as would the K-Ar method.

Most importantly, there is no raw data provided with the manuscript so age calculations cannot be verified or explored. **The manuscript cannot be accepted without this information.**

Specific comments linked to line numbers:

: Use consistent units throughout (Ma).

77: Figure 1 - is low resolution and scale bars cannot be read.

126: Table 2 - Need to express to the precision to the correct number of significant digits, e.g., RHRDV01-4 : 0.42±0.23 Ma.

: Table 2 - Why are the plateau and isochron ages of RJKBR01-h so discordant? Especially considering the total fusion age is also older than the plateau.

141: What does 'first order' mean in this context? Are you implying something about precision or accuracy requirements? I think this is important as it's at the heart of the matter – the balance between the data volume required by provenance studies versus the efforts to obtain the best precision and accuracy with the technique.

151: MSWD should be listed for both age spectra and isochrons. p for isochrons. Ages and uncertainties for isochrons.

151: Since you discuss K/Ca below, it should be illustrated on the plots, along with an indication of the average (integrated) K/Ca that the reader can reference.

: Describe what the gray points are.

: Figures - Resolution is low making figures difficult to read. Needs to be brought up to publication quality. Could resize the isochron diagrams to be the same dimensions as the age spectra without consuming extra page space. The isochrons should be expanded (for example RHRDV01-b, RHRDV01-d or RPJRS01-a), that is, not plotted to the 39Ar/40Ar intercept, but rather to show maximum detail of the data and their relation to the regression line.

Should reconsider the scale on the age spectra. Most of the negative age range is unused and simply compresses the apparent scatter. Some plots, for example RJKBR01-g should be plotted on a finer scale as detail in the spectra cannot be seen.

: "discordant plateau with steps in the same general age range...". A discordant plateau isn't really an age.

: Confusing as written - the ages are not consistent with the proximal terrain, but instead the terrain that borders it. According to your watershed map, the lithic ages should not exceed 5.5 Ma. But this is an important observation; because you've produced age spectra, rather than K-Ar type ages, you can use these reliable dates to ask questions about what transport mechanisms could bring older detritus into this basin. If these were single-step fusions (i.e. K-Ar), you could simply dismiss them as having excess 40Ar. These older clasts effectively highlight why this is a much more robust approach.

: I think you mean to say that K/Ca will be affected by various processes including, but not limited to, ....degree of source partial melting, etc.

: Surely at 1400°C the sample is fused and largely degassed. What does the 37Ar release curve look like?

: This comparison should be with bulk rock data since they would be equivalent to what you measured.

: Or you could simply split the grain, c.f., Ellis et al. 2017
https://doi.org/10.1016/j.chemgeo.2017.03.005

: "This coupled petrological analysis and age determination on single..."

: Interesting point as you would then expect that using a non-atmospheric trapped component (as outlined below) should (could?) bias your ages relative the K-Ar ages in the literature which would have used an atmospheric correction.

: You should show these in the table since you plot them in figure 8.

: Do you mean temperature range? or a singular temperature? multiple steps?

: This should be in the table for reference

: Unclear what's being referred to here - do you mean t(0) regressions? Why would these be different as compared to a standard analysis? Why would the blanks be different? Because this is a single step? Please explain more carefully.

: I'm not at all sure what you're proposing. Are you taking about fusing at a single temperature? Or doing steps only between 680-1140? What happens to the gas from the lower temperature steps? Do you discard it? Have you calculated how much time your method saves relative to a full step-heating experiment, especially as furnaces usually require high temp. burnout between samples?

: Panel B should say "Partial fusion age calculated with..."

: This sample in panel B, RHRDV01-b, becomes even less compatible with the upper age limits indicated by the watershed boundaries (5.5 Ma).

288: You should put an actual 1:1 reference line in here for comparison. The line plotted here has no meaning.

: Fifteen. 4 of them were rejected – at least that is what the table indicates.

Additional comments (recommendations, not requiring action for resubmission):

1) The approach could be significantly improved if the samples were pre-treated more rigorously, i.e., crushed, phenocrysts and alteration phases removed and remaining groundmass leached to obtain pristine groundmass fragments as is usual for basalt dating. Obviously clast sizes limit the amount of datable material that can be processed this way, but this pre-treatment will limit the interfering effects of alteration and excess argon in phenocrysts, making for more robust data. Having a smaller amount of high purity material is better.
2) A CO2 laser-based approach would have lower blanks and can be run 24-7 for greater throughput. But even then, step-heating takes time.
3) I think this approach will struggle to be applied as a high-N type of technique (like U/Pb zircon) because initial conditions and assumptions (initial daughter composition, closed system behaviour) have significant impacts and cannot be easily assessed without detailed analysis.  Single-step analysis will not alleviate this problem.

4) Assuming a 'regional' trapped component composition will only blur the picture of sample ages, giving this technique even less resolution.

5) A better approach would be to pre-screen the materials for key geochemical data and then use the geochemistry as guide to select grains for high precision and accuracy step-heating analysis.  Step heating analysis is the greatest strength of the Ar/Ar method and so should be leveraged for maximum benefit.

---

## Author Comment (AC2)

**Response to Reviewers are shown in Blue below**

Review 1:

We (Brueseke, Benowitz, Trop) found, the currently under discussion manuscript, *New Developments in Incremental Heating Detrital 40Ar/39Ar Lithic (DARL) Geochronology using Icelandic River Sand by* Odinaka Okwueze, Kevin Konrad, and Tomas Capaldi well written and a good contribution to the continued use of the DARL (Detrital Argon Lithics) geochronology approach. We agree the magmatic history of the glaciated Iceland magmatic province will benefit from applications of the DARL technique, as will other relatively remote and glaciated area such as the Cascades Arc of Northwestern United States.

We graciously recommend some key adjustments to the text, given our and others past work doing both 40Ar/39Ar step-heating and modified single grain fusion on ground mass, whole rock chips, and discrete mineral grains from gravel- and sand-sized volcanic-lithic clasts. We first reported 40Ar/39Ar ages on volcanic-lithic grains from modern river sands in the Wrangell Volcanic arc at a 2014 conference (Benowitz et al., 2014), where we demonstrated that a modified heating schedule of sand-sized volcanic lithics was more efficient and accurate for DARL analyses. This was based on incremental heating single sand-sized volcanic-lithic grains and then modifying our fusion schedule based on these results. We also recommended when applying DARL to other regions standard step-heating be performed before developing a fusion or modified (shortened) step-heat schedule. At the time we were concerned about excess 40Ar not excess 36Ar (which Okwueze et al. document). We agree that excess 36Ar is an underappreciated aspect of 40Ar/39Ar geochronology (Benowitz et al., 2018). These method details were explained in a subsequent Geosphere article (Trop et al., 2022; relevant aspects are copied below) and inasmuch, should be noted as where the DARL technique originated and was first published. Furthermore, Kenny et al. (2022) also performed 40Ar/39Ar incremental step-heats on detrital sand volcanic-lithic grains. We also performed and published (Trop et al., 2022) incremental step-heats on volcaniclithic grains when results were questionable or were of key age spans as one of our goals was to determine the age of initiation for the Wrangell Arc. VanderLeest et al (2020) also applied stepheats to detrital clasts.

Thus, we kindly suggest that Okwueze et al. revise their text and clarify that 40Ar/39Ar step-heats and modified fusions were done previously on modern river volcanic-lithic grains, consequently the contribution here builds on these prior studies. This key fact should be made clearer in this manuscript; as-is, the DARL technique as described is not new or particularly novel, especially given that it is centered on n = 15 grains (vs. n = ~2600 grains; Trop et al., 2022). Additionally, Kenny et al., 2022 performed modified step-heats on 50 grains with step counts varying from 2 to15 (?) steps to optimize number of grains vs. diffusion profile information. See their supplemental files.

We understand there is so much literature out there, that it is easy to miss aspects of past research and take no offense and based on conversations with the corresponding author know none was meant. We are genuinely excited to see more DARL work reported from this research team and others.

We thank Brueseke, Benowitz, and Trop for there constructive reviews and appreciate the effort to coordinate thoughts into a single document. I hope no offense was taken in my lack of knowledge on some of the history of the DARL method. In the revised document we have taken time to highlight the past achievements in more detail.

**Specific recommended changes:**
Something like the following for their introduction: *Following previous combination 40Ar/39Ar incremental step-heating and informed modified fusion procedure on modern river volcanic lithic grains (Benowitz et al., 2014, Trop et al., 2022), we developed a new DARL partial fusion procedure specific to the magmatic products of Iceland.*
Below are additional changes and information re: relevant past work we recommend the authors consider during their revision.

We thank you for the thorough review and providing a stronger background of this method. The manuscript has been adjusted to properly cite the background of the method and we clarify the new aspects of the technique we included. We expanded the introduction to capture a more comprehensive review of the method. Furthermore, we defined our contribution as being more focused on low-K glassy volcanics.

"The detrital 40Ar/39Ar lithic (DARL) method is a relatively new detrital geochronological tool that determines the 40Ar/39Ar total fusion or incremental heating age determinations on single grains or multi-grain aliquots recovered from sedimentary deposits (e.g. watersheds) (Benowitz et al., 2014, 2018; Vanderleest et al., 2020; Trop et al., 2022; Kenny et al., 2022). The technique was first reported by Benowitz et al. (2014), wherein incremental heating analyses were undertaken on fine-grained volcanic lithics to propose refined total fusion temperature ranges for rapid DARL analyses. The DARL method was employed to determine the history of the Wrangell Volcanic Arc (Alaska, USA) through 2771 analyses of grains, ranging in size from sand to cobble (Trop et al., 2022). The DARL ages matched the expected age range based on available bedrock analyses (Trop et al., 2022; Brueseke et al., 2023). The chemistry and age results from this technique allowed for novel insights into the evolution of the Wrangell Arc (Alaska, USA) that were only partially observed using traditional U-Pb detrital zircon analyses (Trop et al., 2022; Brueseke et al. 2023). Similarly, Vanderleest et al. (2020) performed incremental heating experiments on igneous clasts separated from a conglomeratic formation (n=7), which provided detrital chronologic constraints on the evolution of the Magallanes-Austral basin within the southern Patagonian Andes. More recently, Kenny et al. (2023) employed the DARL method on 50 sand-sized grained collected from the drainage basin of the sub-glacial Hiawatha impact structure in Greenland. Although none of the grains produced traditionally concordant heating spectrum (e.g. >50% of 39Ar released with more than five consecutive steps), two mini-plateau ages matched resetting ages for detrital zircon. The DARL method has potential limitations due to the lower closure temperatures of Ar and greater susceptibility of age disturbances due to alteration as compared to the detrital zircon method. However, in environments that contained mixed mafic and felsic lithologies (e.g. volcanic arcs) or consist primarily of fine-grain extrusive volcanics (e.g. Iceland), the DARL method allows for novel insights not obtainable by the traditional detrital mineral phases. Here we expand upon the method through incremental heating experiments on

**single coarse sand or fine gravel grains of volcanic lithic fragments from Icelandic rivers. These sedimentary deposits primarily consist of glassy or fine-grained low-K mafic lava flows and if ages can be reliably constrained with the DARL method, then other low-zircon fertility terrains such as arc and intraplate ocean islands can be constrained. Based on the incremental heating results we propose a methodology for rapid fusion analyses of glass-rich volcanic lithics."**

**Around line 15** (Benowitz et al., 2014; VanderLeest et al., 2020; Kenny et al., 2022; Trop et al., 2022 did 40Ar/39Ar incremental step-heats on detrital cobbles and/or sand). Here we present a new methodology for capturing the magmatic history of fine grained extrusive volcanic rocks using single grain detrital 40Ar/39Ar incremental heating geochronology. The DARL (or Detrital Argon Lithics) **method thus far** has consisted of 40Ar/39Ar total fusion analyses, which pose a problem in the case of Iceland, due to the nature of its young glassy lava flows commonly displaying subatmospheric 40Ar/36Ar isochron intercepts and low 40Ar*.

**Changed to: "The DARL (or Detrital Argon Lithics) method has consisted of $^{40}$Ar/$^{39}$Ar incremental heating and total fusion analyses, which has not yet been applied to predominantly mafic terrains composed of young glassy lava flows, which commonly display subatmospheric $^{40}$Ar/$^{36}$Ar isochron intercepts and low $^{40}$Ar*."**

**Around line 25** Benowitz et al., 2014**;** Trop et al., 2022 did both a combination of informed single grain fusions based on incremental heating results; n = ~2600 grains are what was eventually analyzed and reported in Trop et al (2022) For this reason, **we propose combining the aspects of the total fusion and incremental heating** DARL methodologies to acquire age data for the large N values needed for detrital studies while improving the accuracy of total fusion DARL analysis.

Changed to: "For this reason, we build upon a previously proposed method that combines total fusion and incremental heating DARL methodologies to acquire age data for the large N values needed for detrital studies of mafic volcanic terrains."

**Around line 40 (**DARL has been applied to sand and pebble grains and cobbles, and as a combination of modified fusion and incremental step-heating…. Benowitz et al., 2014; VanderLeest et al 2020; Kenny et al., 2022; Trop et al., 2022**)**
The detrital 40Ar/39Ar lithic (DARL) method is a relatively new detrital geochronological tool that **thus far** employed 40Ar/39Ar total fusion analyses on single grains or multi-grain aliquots recovered from cobble sized (>10 cm) volcanic sediments (Trop et al., 2022; Brueseke et al., 2023).

**Changed to: "The detrital $^{40}$Ar/$^{39}$Ar lithic (DARL) method is a relatively new detrital geochronological tool that determines the $^{40}$Ar/$^{39}$Ar total fusion or incremental heating age determinations on single grains or multi-grain aliquots recovered from sedimentary deposits (e.g. watersheds) (Benowitz et al., 2014, 2018; Vanderleest et al., 2020; Trop et al., 2022; Kenny et al., 2022)."**

**Around line 50 (**this has already been done…Benowitz et al., 2014; Kenny et al., 2022; Trop et

al., 2022**)** Here **we expand** upon the method through incremental heating experiments on single coarse sand grains of volcanic lithic fragments from Icelandic rivers.

**This section of the paragraph was changed considerably to reflect this, see first comment response above.**

**Line 220 (**This seems a little overstated given ~10 years of DARL step-heating work and the orders of magnitude larger number of individual DARL analyses from Trop et al., 2022 and the combination of geochemistry and DARL dating in VanderLeest et al 2020 and Brueseke et al., 2023).

Provided the level of difficulty, the incremental heating DARL experiments worked well and **represent an advancement in the field of detrital geochronology.**

**Sentence removed for simplicity.**

**Around line 250** (this was sort of done -Trop et al., 2022- to evaluate alteration and excess 40Ar and for sure the DARL method has been applied to dominantly mafic bedrock sources.) Although the internal concordance test afforded by the incremental heating method has many advantages, the long analyses time hinders the method's use for detrital geochronology studies, which rely on high= N values. Therefore, **we propose that a subset of grains from a sampling site be analyzed with the incremental heating method** in order to define the best partial fusion temperature ranges and appropriate assumed 40Ar/36Ar0. More work is required to assess the validity of the method in different geologic settings, but the primary data from this study indicates the method is valid and **allows for detrital geochronology studies of dominantly mafic bedrock sources.**

**Changed to: "**A single incremental heating experiment using a vacuum furnace takes ~12 hours to complete. Therefore, a rapid analyses method is required to obtain the large N values needed for a successful detrital geochronology study. Trop et al. (2022) used incremental heating on a subset of grains to assess for alteration or excess argon. Thereafter, they employed the total fusion method wherein individual grains or multi-grain aliquots were fused in a single step (Trop et al., 2022). An atmospheric $^{40}Ar/^{36}Ar_0$ was assumed with the age calculations and the results were ~equivalent to K/Ar ages collected from the region."

**Around Line 260** (at the time we used 295.5 for atmospheric 40Ar/36Ar0)…which now is not standard…but does make the reference to our work a bit confusing…perhaps remove?). An atmospheric 40Ar/36Ar0 was assumed with the age calculations and the results were ~equivalent to K/Ar ages collected from the region.

That shouldn't make a significant difference as the atmospheric to subatmospheric shift will be the same degree but show lower values since your mass discrimination factors on air standards were normalized to 295.5 instead of 298.6. E.g. this Iceland sample set would have a 40Ar/36Ar0 mean around 292 if I used 295.5 for my MDF corrections.

**Around line 295**
The DARL method provides a novel means of constraining the volcanic history of a region through detrital geochronology of lithic grain sand samples.
Please Add the reference to Trop et al. (2022), given that is where the DARL technique originated and was first published:
The DARL method (***Trop et al., 2022***) provides a novel means of constraining the volcanic history of a region through detrital geochronology of lithic grain sand samples.

Changed to: "The DARL method (Trop et al., 2022) provides a novel means of constraining the volcanic history of a region through detrital geochronology of lithic grain sand samples."

**Other manuscript notes that need to be addressed:**
Please define what you mean by discordant: We think we know what you are referring, but it is never defined/explained how you are applying this broad term.

Added to the methods (~line 122): "We define a successful age plateau as containing five or more consecutive heating steps that incorporate over 50% of $^{39}Ar_K$ and have a probability of fit factor >5%. If a heating step is not within uncertainty of the plateau than we refer to that as a discordant step."

Table 1: Please add the known age range for magmatism for each sample/drainage.

Added. Caption updated: **Table 1:** "The location and general geomorphology of each sampling site location. Age ranges are approximated from available outcrop $^{40}Ar/^{39}Ar$ and K/Ar age determinations collated in Jóhannesson and Sæmundsson (2009). "

How often did you measure mass discrimination? Did it drift? Could applying the "incorrect" mass discrimination explain your excess 36Ar (and excess 40Ar) measurements? 36Ar was measured on a more sensitive electron multiplier? Where 40Ar was measured on a sensitive (but less so?) faraday? Is this a factor in the excess 36Ar measurements?

We doubt these are controlling factors on the excess 36Ar measurements…but these factors should be at least documented and mentioned-dismissed in the text/methods.

Some of these factors were mentioned in the methods and the raw data is provided in the new supplements. We have adjusted the methods to expand on these calibration procedures. "Five air standards (for mass discrimination factors; MDF) and collector calibrations (for faraday-ion multiplier calibration) were run prior to every experiment. The MDF (assuming a 40Ar/36Aratmo = 298.56 ± 0.31; Lee et al. 2006) and calibration factors for an individual experiment were determined by fitting a polynomial curve to the results over two weeks and interpolating the values for when the experiment was run. Collector calibrations are done by putting 36Ar (from air) on the multiplier then on the faraday by adjusting the magnet. This is repeated 75 times per analyses to determine the multiplier/faraday offset. Five collector calibrations were run per day (immediately after the MDF analyses before pumping the air out of

the mass spec). Neither the collector calibrations nor MDF results varied significantly over the course of the project"

It also important to note that other samples (not related to this project) run during the same time consistently produced atmospheric intercepts so I believe the phenomenon is related to the geology of the samples as opposed to analytical factors.

Perhaps more discussion on how modern mass spectrometer instrumentation allows for the clearer identification of excess 36Ar could be added?

Although this could be valuable I do not see a clear location to include this discussion in the manuscript.

**Around Line 275**
Therefore, we can calculate the partial-fusion age between those temperature steps, using an 40Ar/36Ar0 that is representative of our dataset (296 ± 4; Figure 8).

What was the range of determined 40Ar/36Ar0 for all the grains analyzed?
289.7 to 300.3….Is it really sensible to assume a single subatmospheric 40Ar/36Ar0 for all samples?

The justification for using a value of 296 +/- 4 is that the larger uncertainty covers the range from subatmospheric to slightly supra atmospheric. The nature of the TF DARL method comes with inherent uncertainties (not unlike detrital zircon) but the high precision at which the atmospheric value is known (+/- 0.31) is clearly too precise to cover the natural range observed in these young volcanic products. However, if we employ the total range (287.1 – 304.3 or 295.7 +- 8.6) then each age uncertainty will become unusable.

Given most results approximated or were greater than 298.56 ± 0.62 (Lee et al., 2006)? The 296 ± 4: Is that a weighted average? The uncertainty is propagated during the age calculations?

Correct, the 296 is from the weighted average. The +- 4 uncertainty was propagated during the 'partial fusion' calculations, hence the larger uncertainties throughout. Those values are now provided in Table 3.

**Isochron plots:**
Are the same steps used for the plateau age determinations used for the isochron age determinations? They should be. It seems for some of the samples this is not the case? It is hard to tell given the number of steps used in the isochron determinations are not listed in table 2. If always the same number of steps/same steps are used for isochron regressions as were used for the plateau age determinations (as they should be? Unless justified), please mention in text.

**In all scenarios the isochron points and plateau points are the same heating steps. This was addressed in the methods "**When a sample contained a concordant isochron with a non-atmospheric $^{40}Ar/^{36}Ar_0$ intercept (following the same statistical criteria as the described for the plateau), the plateau was recalculated using the intercept and uncertainty (e.g. Heaton and

Koppers, 2019). When a plateau was recalculated, no additional heating steps were added —
even if they became concordant due to the increased intercept uncertainty."

We added the following line to the table 2 caption for clarity:
"n=heating steps used in age calculations for both plateau and isochron"

**RPJSO1-e**
Would you consider this stepping up age spectrum indicative of loss? If so, is it appropriate to
perform a regression back to initial 40Ar/36Ar (Isochron plot) given the documented loss?

**I would consider this result indicative of partial degassing (which is common in terrains
with overlapping lava flows). However, the concordant plateau at higher temperature (and
corresponding isochron) is perfectly reasonable within standard 40Ar/36Ar practice. The
atmospheric regression at mid-high temperature is further evidence this sample formed in
equilibrium with atmosphere. There is no reason to remove this isochron.**

**Figure 10**
This is a key figure…but we don't see the negative original age determinations in Table 2 and
there are no supplemental isotopic files. Please add the negative (original) age determinations to
Table 2 and add full supplemental files. Schaen et al. (2021) community based (dozens of noble
gas lab authors) makes a strong case and sets out examples of how 40Ar/39Ar isotopic
information should be documented in scientific manuscripts. Regardless if Schaen et al. (2021) is
followed to the "T", detailed isotopic tables are required to be included with 40Ar/39Ar
geochronology publications to be able to evaluate the authors results/interpretations/methods.

**We acknowledge that the omission of the original full of set of isotopic information was a
mistake. We have appended a supplemental document to the manuscript that includes full
information suites for all analyses as individual tabs in an excel file. We also added Table 3
to show the age information used in Figure 10.**

| Sample | Incremental Heating Results | | Total Fusion Results | | Total Fusion (680 to 1140°C) | |
|---|---|---|---|---|---|---|
| | Age (Ma) | ± 2σ (i) | Age | ± 2σ (i) | Age | ± 2σ (i) |
| RSTDR01-a | 13.4 | 0.3 | 14.6 | 0.3 | 14.1 | 0.3 |
| RSTDR01-d | 13.5 | 0.2 | 13.9 | 0.4 | 13.8 | 0.6 |
| RHRDV01-a | 7.5 | 0.4 | 6.5 | 0.6 | 6.3 | 1.4 |
| RHRDV01-b | 8.6 | 0.1 | 8.9 | 0.3 | 11.2 | 2.2 |
| RHRDV01-d | 0.4 | 0.2 | -2.3 | 0.3 | 1.0 | 0.4 |
| RJKBR01-a | 0.5 | 0.2 | -0.7 | 0.5 | 1.8 | 1.7 |
| RJKBR01-g | 2.04 | 0.02 | 1.97 | 0.04 | 2.04 | 0.04 |
| RJKBR01-h | 1.8 | 0.8 | 3.7 | 0.7 | 5.2 | 1.5 |
| RJKBR01-k | 0.5 | 0.4 | 0.6 | 0.6 | 0.7 | 0.7 |
| RSTLK01-a | 10.7 | 0.2 | 11.0 | 0.2 | 10.6 | 0.3 |
| RPJRS01-a | 0.3 | 0.1 | 0.7 | 0.1 | 0.2 | 0.2 |
| RPJRS01-b | 0.6 | 0.2 | 1.8 | 0.4 | 0.4 | 0.8 |
| RPJRS01-c | 0.6 | 0.1 | 0.7 | 0.1 | 0.7 | 0.2 |
| RPJRS01-d | 0.4 | 0.1 | 0.6 | 0.2 | 0.8 | 0.3 |
| RPJRS01-e | 0.2 | 0.2 | -1.3 | 0.2 | 0.3 | 0.7 |

**Table 3:** A comparison of concordant plateau age, total fusion ages assuming all gas released and an atmospheric $^{40}Ar/^{36}Ar_0$ value and total fusion results for steps between 680° and 1140°C assuming a $^{40}Ar/^{36}Ar_0$ of 296 ± 4. Steps with a discordant heating spectrum are excluded from the table.

We see on table 2 you correct for excess 36Ar, but don't correct for excess 40Ar. Would it be better to use the original isochron age determinations for all analysis instead of plateau ages?

**The corresponding ages when we correct for a non-atmospheric $^{40}Ar/^{36}Ar_0$ and if we use the isochron ages are the same with slightly lower errors for the plateau (as a function of the weighted mean calculation). For simplicity for the non-argon geochronologist reader we prefer to just recalculate the plateau age in order to reduce the likelihood of someone using the wrong age.**

Figure 10….B looks stretched? i.e., Why are uncertainties so big?
Or are uncertainties blown up with the applied 296 ± 4 40Ar/36Ar0, hence MSWD goes down simply because of the larger uncertainties? Compared to 298.56 ± 0.62 40Ar/36Ar0 (Lee et al., 2006).

**Correct, the samples are recalculated with a larger $^{40}Ar/^{36}Ar_0$ intercept uncertainty, which results in large age uncertainties. One advantage with this method is that the 'geologic' uncertainty on the total fusion age is much larger compared to the analytical uncertainty. Using this larger intercept values makes the single age determinations uncertainties more realistic given the nature of the method.**

What would be the MSWD for graph A be if the youngest three ages were parsed? Seems those are biasing everything and for graph B all the ages are being modified (some far away from there "actual ages"!!!).

**The modification from the actual ages is a key feature of this method. Although some of the samples will have their absolute ages become less accurate, the larger uncertainties account for it and allow for the other ages to fall in line. The goal is to generate a more accurate dataset at the cost of precision. For panel A, the MSWD stays high if the youngest ages are removed due to other the two large age offsets.**

**Can you please add a table** of original ages/uncertainties for all samples vs. modified ages/uncertainty with the assumed $296 \pm 4$ $40Ar/36Ar_0$ determination. We think this is a key aspect…Yes you are shifting the youngest ages, but you are also shifting the other ages, Is that appropriate given the large variations in actual measured/calculated $40Ar/36Ar_0$?

**Table 3 added. See above.**

For example, on figure 10….sample RHDRV01-b gets shifted from a total fusion age of 8.9 Ma ($\pm 0.03$) with a small uncertainty and becomes >11.0 Ma on 10B with a huge uncertainty ($\pm \sim 5$ Ma?).

Is this an improvement over the original accurate and precise age determinations?

**The age shifts from $8.9 \pm 0.3$ Ma to $11.2 \pm 2.2$ Ma. Although the total fusion age and plateau age are within uncertainty of each other there would be no way to know this if only a TF age was calculated. Therefore, this method shirks precision for the likelihood of increased accuracy (through more conservative uncertainties). This will always be a trade off with the DARL method.**

Can you get negative ages simply due to statistics? i.e. An age result of 10 ka $\pm$ 20 ka on a lava means given enough analyses you would get a negative age from the same sample.
We are not sure if trying to make "exact" geologic interpretations from modified negative $40Ar/39Ar$ follows best practices. Yes these grains are young and the authors can robustly state that, but we are not sure applying a $296 \pm 4$ $40Ar/36Ar_0$ to a negative age with a measured 289.67 $40Ar/36Ar_0$ makes for a geologically more meaningful age.

**I see what you are saying about the negative ages due to statistics but would counter that the negative ages in our case have statistically concordant isochrons. As for whether a mean of the measured intercept values should be used or the full range will be a difficult question for future researchers to decide.**

Rough figure showing large shift from measured to modeled ages.
Is 1.79 Ma age on Figure 10?
(RJKBR01-h)? might be….
forgive us if it is.

**Yes, it the large offset right above the young sub-atmospheric cluster on the figure.**

**Line 80** (>2 mm sized grains are granule sized gravel as opposed to sand sized grains, so the text should state that fine gravel (or granules) and sand was analyzed**).**

The bulk sediment samples were sieved and grains from the **2-3 mm size fraction** were selected for all sites except RJKBR01, where the 1-2 mm size fraction was used. Each selected grain was separated and given a unique identifier (i.e. -A; Figure 2).

**Thanks for the clarity. Sands changed to sands/fine gravel throughout.**

**Data Availability**
Please include a link to all isotopic information (preferably in excel format) and supplemental figures using a file-sharing site like https://zenodo.org/records/802100. As is, it is impossible to replot the presented data, evaluate the results, etc.

**All data made available through the supplement now.**

**Summary Suggestion:**
Perhaps a better DARL method for Iceland would be to: Degass/not measure/pump out lower temperature steps (below 680 °C). And then a apply a 296 ± 4 40Ar/36Ar0 for the negative age determinations: but acknowledge these modeled age determinations are approximations and not indicative of exact geological eruptive events.

**This is a good suggestion and the following line was added to the discussion: "Alternatively, since the sensitivity to the sub atmospheric intercepts seems greater in the youngest samples, perhaps the alternate 40Ar/36Ar0 (296 ± 4) should only be used when a sample produces a negative age result."**

**Review References:**
Benowitz, J.A., Davis, K.N., Brueseke, M.E., Trop, J.M., and Layer, P., 2014, Investigating the lost arc: Geological constraints on ~25 Million years of magmatism along an arc-transform junction, Wrangell Volcanic Belt, Alaska, Geological Society of America Abstracts with Programs, Vol. 46, No.6, p.363.
Benowitz, J.A., Miggins, D.P., Koppers, A.A. and Layer, P.W., 2018, November. Why are some young volcanic rocks undateable: Chemistry, Environment, or instrumentation?. In *GSA Annual Meeting in Indianapolis, Indiana, USA-2018*. GSA.
Kenny GG, Hyde WR, Storey M, Garde AA, Whitehouse MJ, Beck P, Johansson L, Sondergaard AS, Bjork AA, MacGregor JA, Khan SA. 2022, A Late Paleocene age for Greenland's Hiawatha impact structure. Science Advances.
Schaen, A.J., Jicha, B.R., Hodges, K.V., Vermeesch, P., Stelten, M.E., Mercer, C.M., Phillips, D.,
Rivera, T.A., Jourdan, F., Matchan, E.L. and Hemming, S.R., 2021. Interpreting and reporting 40Ar/39Ar geochronologic data. *GSA Bulletin*, *133*(3-4), pp.461-487
Trop, J.M., Benowitz, J.A., Kirby, C.S. and Brueseke, M.E., 2022. Geochronology of the Wrangell
Arc: Spatial-temporal evolution of slab-edge magmatism along a flat-slab, subduction-transform transition, Alaska-Yukon. *Geosphere*, *18*(1), pp.19-48.

VanderLeest, R.A., Fosdick, J.C., Leonard, J.S. and Morgan, L.E., 2020. Detrital record of the late
Oligocene–early Miocene mafic volcanic arc in the southern Patagonian Andes (~ 51° S) from single-clast geochronology and trace element geochemistry. *Journal of Geodynamics, 138*, p.101751.

---

## Author Comment (AC3)

**Response to Reviewers are shown in Blue below**

**Review #2:** New Developments in Incremental Heating Detrital 40Ar/39Ar Lithic (DARL) Geochronology using Icelandic River Sand, Okwueze et al. (2024).

Overview :

This contribution attempts to improve on the DARL method by overcoming the inherent limitations in previous applications that employed exclusively K-Ar methods. This is done using a detailed 40Ar/39Ar step heating approach. Much of the data are of high quality and show excellent release spectra. Data from some of the more complex step heating spectra are reasonably discussed.

Application of non-atmospheric trapped compositions for correcting plateau ages is also explained and justified. However, in the case of apparent non-atmospheric trapped components, I would use the isochron ages as these will be less affected by the trapped component issues.

The proposed method of 'partial fusion+averaged trapped component' is very poorly explained and it's implications are not clear. As read in the text, it appears to give a different, but similarly blurred picture of the age distribution of clastic materials as would the K-Ar method.

Most importantly, there is no raw data provided with the manuscript so age calculations cannot be verified or explored. **The manuscript cannot be accepted without this information.**

**We thank you for your detailed review and expertise. The lack of supplements on the original submission was a major oversight and we have corrected it here.**

Specific comments linked to line numbers:

: Use consistent units throughout (Ma).

**All units switched to Ma.**

77: Figure 1 - is low resolution and scale bars cannot be read.

**Figure 1 edited slightly and exported at higher resolution.**

126: Table 2 - Need to express to the precision to the correct number of significant digits, e.g., RHRDV01-4 : 0.42±0.23 Ma.

**All significant digits fixed throughout text, figures and tables.**

: Table 2 - Why are the plateau and isochron ages of RJKBR01-h so discordant? Especially considering the total fusion age is also older than the plateau.

**This sample has very high atmosphere/radiogenic ratio (<3% radiogenic 40Ar % per step) and as such the isochron clusters near the atmosphere origin. Thus the isochron age has significant scatter. The total fusion age from this sample incorporates the highest temperature steps, which had excess argon and correspondingly much older apparent ages**

**( up to 14 Ma). However, the plateau still appears to be concordant with an atmospheric $^{40}Ar/^{36}Ar_0$ is employed, therefore the plateau age is preferred.**

141: What does 'first order' mean in this context? Are you implying something about precision or accuracy requirements? I think this is important as it's at the heart of the matter – the balance between the data volume required by provenance studies versus the efforts to obtain the best precision and accuracy with the technique.

**I agree that defining first order is important. I will also admit I overuse the term. I removed it in a few places. On line 252 I clarify that first order means lower precision: "which can provide a first order (e.g. low precision) assessment"**

151: MSWD should be listed for both age spectra and isochrons. p for isochrons. Ages and uncertainties for isochrons.

**Table 2 now has the MSWD, P, ages and uncertainties for spectra and isochrons listed. This information is also in the new supplemental document.**

151: Since you discuss K/Ca below, it should be illustrated on the plots, along with an indication of the average (integrated) K/Ca that the reader can reference.

**Added to each of the age determination result figures.**

: Describe what the gray points are.

**Added ", grey squares are excluded steps."**

: Figures - Resolution is low making figures difficult to read. Needs to be brought up to publication quality. Could resize the isochron diagrams to be the same dimensions as the age spectra without consuming extra page space. The isochrons should be expanded (for example RHRDV01-b, RHRDV01-d or RPJRS01-a), that is, not plotted to the 39Ar/40Ar intercept, but rather to show maximum detail of the data and their relation to the regression line.

Should reconsider the scale on the age spectra. Most of the negative age range is unused and simply compresses the apparent scatter. Some plots, for example RJKBR01-g should be plotted on a finer scale as detail in the spectra cannot be seen.

**All age determination results were cleaned up to increase clarity and add the desired K/Ca results. The apparent age scale bar was kept the same as -20 to 20 Ma to maintain clear comparison between the results. The supplemental document has all the necessary plots and the axes can be manipulated therein.**

: "discordant plateau with steps in the same general age range...". A discordant plateau isn't really an age.

**This is a fair point and the sentence was modified: "Out of the three grains that were analysed from the RSTDR01 sample (Stadará River), two produced concordant plateaus at ca. 13.4 Ma, and one produced a discordant plateau (Figure 3)."**

: Confusing as written - the ages are not consistent with the proximal terrain, but instead the terrain that borders it. According to your watershed map, the lithic ages should not exceed 5.5 Ma. But this is an important observation; because you've produced age spectra, rather than K-Ar type ages, you can use these reliable dates to ask questions about what transport mechanisms could bring older detritus into this basin. If these were single-step fusions (i.e. K-Ar), you could simply dismiss them as having excess 40Ar. These older clasts effectively highlight why this is a much more robust approach.

**This is an excellent observation. I have modified the sentence accordingly to capture your suggestion: "The RHRDV01 samples (Heradsvötn River) had three of four successful age determinations with two older (7.5 and 8.6 Ma) and one young (0.42 Ma) results (Figure 4). The older ages are not consistent with the currently proximal Pliocene-upper Miocene volcanic bedrock terrain (3.3-5.5 Ma) or the younger terrain upstream. These unexpected results highlight a strength of the incremental heating DARL method. Having the ability to obtain more robust age spectrums than traditional K/Ar or total fusion methods allows for more detail questions to be asked on sediment transport mechanisms as opposed to simply discarding unexpected outliers assuming excess 40Ar."**

: I think you mean to say that K/Ca will be affected by various processes including, but not limited to, ....degree of source partial melting, etc.

**Fixed accordingly: "The $^{40}$Ar/$^{39}$Ar method provides a means of assessing the ratio of K (through $^{39}$Ar$_K$ proxy) to Ca ($^{37}$Ar$_{Ca}$), which can provide a first order (e.g. low precision) assessment of a variety of processes, including but not limited to, the degree of source melt enrichment, degree of mantle melting or the assimilation/crystal fractionation history of the sample."**

: Surely at 1400°C the sample is fused and largely degassed. What does the 37Ar release curve look like?

**Release curves added to figures 3 - 7. Some $^{37}$Ar will keep being released at higher temperatures based on some unpublished experiments I've done. The 39Ar and 40Ar* are essentially done at 1400C.**

: This comparison should be with bulk rock data since they would be equivalent to what you measured.

**Fixed. The figure was modified accordingly. Figure caption: "Figure 9: The bulk rock K/Ca and alkali index values for known Icelandic volcanics. The K/Ca values for the analysed grains are shown as red bars on the y-axis. Icelandic sample data from the compiled GEOROC database (DIGIS Team, 2023) and filtered to K/Ca <1 to remove rhyolite samples. The alkali index is calculated as [Na2O + K2O] – [SiO2 x 0.369 - 14.350] (Rhodes et al., 2012)."**

: Or you could simply split the grain, c.f., Ellis et al. 2017
https://doi.org/10.1016/j.chemgeo.2017.03.005

**Option added. "Additional non-destructive (e.g. scanning electron microscopy [SEM] analyses), split grain (e.g. Ellis et al., 2017) or semi-destructive (e.g. laser ablation – inductively coupled plasma mass spectrometry [LA-ICPMS]) analyses prior to irradiation would be required to more thoroughly trace petrologic evolution"**

: "This coupled petrological analysis and age determination on single..."

**Fixed as suggested: "This coupled petrologic analysis and age determination on single grains would provide novel insights into the long-term first-order evolution of a volcanic terrain."**

: Interesting point as you would then expect that using a non-atmospheric trapped component (as outlined below) should (could?) bias your ages relative the K-Ar ages in the literature which would have used an atmospheric correction.

**Correct.**

: You should show these in the table since you plot them in figure 8.

**Added Table 3**

: Do you mean temperature range? or a singular temperature? multiple steps?

**Changed to "…temperature range"**

: This should be in the table for reference

**Added with Table 3**

: Unclear what's being referred to here - do you mean t(0) regressions? Why would these be different as compared to a standard analysis? Why would the blanks be different? Because this is a single step? Please explain more carefully.

**Expanded the sentence to further explain: "It is important to note that the partial fusion errors here are overestimated due to expected lower blank corrections (using a single preceding blank instead of a polynomial fit to multiple blanks) and peak regression uncertainties (higher peak signals released in a single heating step) during an actual fusion measurement."**

: I'm not at all sure what you're proposing. Are you taking about fusing at a single temperature? Or doing steps only between 680-1140? What happens to the gas from the lower temperature steps? Do you discard it? Have you calculated how much time your method saves relative to a full step-heating experiment, especially as furnaces usually require high temp. burnout between samples?

**To increase clarity on the recommended method we added the following:**

**"Based on these preliminary Icelandic results, our next recommended steps would be to pre-heat the grains to 680°C while under active vacuum, then perform a single 1140°C heating step to obtain the age of the grain. This will allow for 100s of grains to be analysed within a reasonable timeframe — providing the large N values needed for a detrital geochronology experiment. Alternatively, since the sensitivity to the sub atmospheric intercepts seems greater in the youngest samples, perhaps the alternate $40Ar/36Ar_0$ (296 ± 4) should only be used when a sample produces a negative age result."**

: Panel B should say "Partial fusion age calculated with..."

**Good catch. Fixed.**

: This sample in panel B, RHRDV01-b, becomes even less compatible with the upper age limits indicated by the watershed boundaries (5.5 Ma).

**I concur the absolute age puts it further away but the larger uncertainties still remain within error of the original age.**

288: You should put an actual 1:1 reference line in here for comparison. The line plotted here has no meaning.

**A dashed line was added as a reference 1:1 line. The best fit line is still included as I believe it helps show that the partial fusion ages are a closer fit to the expected 1:1 result.**

: Fifteen. 4 of them were rejected – at least that is what the table indicates.

**Changed to "15 of 19"**

Additional comments (recommendations, not requiring action for resubmission):

1) The approach could be significantly improved if the samples were pre-treated more rigorously, i.e., crushed, phenocrysts and alteration phases removed and remaining groundmass leached to obtain pristine groundmass fragments as is usual for basalt dating. Obviously clast sizes limit the amount of datable material that can be processed this way, but this pre-treatment will limit the interfering effects of alteration and excess argon in phenocrysts, making for more robust data. Having a smaller amount of high purity material is better.

**I agree but the balance between mass available (e.g. 5-8mg per 2mm grain) and the age/nature of the samples (young basalts) make losing material difficult. We considered doing this on larger grains in the deposit but then would bias the results towards proximal deposits as lava flows are not robust like zircons during transport. The method definitely needs more refining and we had plans to run a series of variable acid leaching experiments on individual grains from the same lava flows (at variable sieve sizes) to obtain a best pretreatment plan. However, the nature of this project (undergraduate thesis) didn't allow the time needed. I appreciate the recommendations for future investigations.**

2) A $CO_2$ laser-based approach would have lower blanks and can be run 24-7 for greater throughput. But even then, step-heating takes time.

**I agree and that is the next step. The concern originally with using the laser was the grains are rounded and the laser will unevenly heat the large single masses. However, this is likely a minor effect (if any) and next we will use the $CO_2$ laser.**

3) I think this approach will struggle to be applied as a high-N type of technique (like U/Pb zircon) because initial conditions and assumptions (initial daughter composition, closed system behaviour) have significant impacts and cannot easily assessed without detailed analysis. Single-step analysis will not alleviate this problem.

**Although true, the alternative at this time is there are no options for detrital studies in mafic terrains. I believe some data with caveats is better than no attempts.**

4) Assuming a 'regional' trapped component composition will only blur the picture of sample ages, giving this technique even less resolution.

**As mentioned a few times in this response, the balance between accuracy and precision is key here. I rather have lower precision but a better chance of accuracy. If the we are dealing with a terrain like Iceland that has 15 m.y. of history then 0.5-1Ma uncertainties aren't the worst. But there is definitely room for improvement still. I also added the suggested alternative of using an alternative trapped component only for negative ages in the discussion.**

5) A better approach would be to pre-screen the materials for key geochemical data and then use the geochemistry as guide to select grains for high precision and accuracy step-heating analysis. Step heating analysis is the greatest strength of the Ar/Ar method and so should be leveraged for maximum benefit.

**This is ideal and suggested as part of the discussion. Screening is always a bit tricky as you have the potential to bias your results against certain lithologies that aren't ideal for 40Ar/39Ar (e.g. olivine rich rocks) but perhaps a defined set of 'red flags' that require the grain be step heated instead of fused could be employed.**